# Tapasin assembly surveillance by the RNF185/Membralin ubiquitin ligase complex regulates MHC-I surface expression

Michael L. van de Weijer [1], Krishna Samanta[1,8], Nikita Sergejevs [1,8], LuLin Jiang[2,5], Maria Emilia Dueñas [3,6], Tiaan Heunis[1,7], Timothy Y. Huang [2], Randal J. Kaufman [2], Matthias Trost [3], Sumana Sanyal [1], Sally A. Cowley [1,4] & Pedro Carvalho [1] ✉

Immune surveillance by cytotoxic T cells eliminates tumor cells and cells infected by intracellular pathogens. This process relies on the presentation of antigenic peptides by Major Histocompatibility Complex class I (MHC-I) at the cell surface. The loading of these peptides onto MHC-I depends on the peptide loading complex (PLC) at the endoplasmic reticulum (ER). Here, we uncovered that MHC-I antigen presentation is regulated by ER-associated degradation (ERAD), a protein quality control process essential to clear misfolded and unassembled proteins. An unbiased proteomics screen identified the PLC component Tapasin, essential for peptide loading onto MHC-I, as a substrate of the RNF185/Membralin ERAD complex. Loss of RNF185/Membralin resulted in elevated Tapasin steady state levels and increased MHC-I at the surface of professional antigen presenting cells. We further show that RNF185/Membralin ERAD complex recognizes unassembled Tapasin and limits its incorporation into PLC. These findings establish a novel mechanism controlling antigen presentation and suggest RNF185/Membralin as a potential therapeutic target to modulate immune surveillance.

The biogenesis of secretory and membrane proteins in the endoplasmic reticulum (ER) is monitored by ER-associated degradation (ERAD)[1,2]. This conserved quality control system detects misfolded, unassembled, and mislocalized proteins and promotes their transport back to the cytosol for degradation by the proteasome. ERAD also targets some folded proteins in a signal-dependent manner, such as specific enzymes involved in sterol biosynthesis, thereby playing central roles in both protein and lipid homeostasis[1,3].

Mechanistically, ERAD is carried out by a variety of ubiquitin ligase complexes integral to the ER membrane, each with specificity for different classes of substrates[4]. In all cases, subunits of ERAD complexes recognise substrates in the lumen or membrane of the ER and facilitate their movement across the ER membrane toward the cytosol for ubiquitination. All ERAD complexes converge on the cytosolic p97 ATPase complex that pulls ubiquitinated substrates from the membrane and hands them to the proteasome for degradation[2,5].

An ERAD complex consisting of the ubiquitin ligase RNF185, the multispanning membrane protein Membralin (MBRL), and a member of the TMUB family—either TMUB1 or TMUB2—was recently identified[6,7]. We previously showed that this RNF185/MBRL complex

[1]Sir William Dunn School of Pathology, University of Oxford, South Parks Road, Oxford OX1 3RE, UK. [2]Degenerative Diseases Program, Genetics, and Aging Research Center, Sanford Burnham Prebys Medical Discovery Institute, La Jolla, CA 92037, USA. [3]Biosciences Institute, Newcastle University, Framlington Place, Newcastle upon Tyne NE2 4HH, UK. [4]James and Lillian Martin Centre for Stem Cell Research, Sir William Dunn School of Pathology, University of Oxford, Oxford, UK. [5]Present address: Altos Labs-Bay Institute of Science, Redwood City, CA, USA. [6]Present address: Telethon Kids Institute, Perth, Nedlands, WA 6009, Australia. [7]Present address: Immunocore Ltd, 92 Park Drive, Abingdon OX14 4RY, UK. [8]These authors contributed equally: Krishna Samanta, Nikita Sergejevs. ✉e-mail: pedro.carvalho@path.ox.ac.uk

promotes the degradation of a subset of membrane proteins[7]. However, the complete set of RNF185/MBRL substrates remains unknown. Interestingly, genetic studies showed that MBRL ablation is perinatal lethal in mice[8]. While indistinguishable from WT littermates at birth, MBRL KO mice display acute loss of motor neurons and hyperreactive astrocytes, resulting in death around day 5. Astrocyte-specific ablation of MBRL resulted in a similar albeit milder phenotype, with the lethality occurring around day 25[9]. Despite being ubiquitously expressed, the lethality of MBRL deletion was rescued by expression of a neuronal MBRL transgene, indicating an essential role for this protein in the central nervous system, perhaps in astrocytes. How the severe mouse phenotype relates to MBRL function in ERAD is unclear.

Among the complexes assembled in the ER of vertebrate cells is the peptide loading complex (PLC). The PLC transports antigenic peptides generated in the cytosol into the ER lumen and loads them onto major histocompatibility complex class I (MHC-I) molecules. Once loaded, MHC-I molecules traffic to the plasma membrane and display the peptides at the cell surface for immune recognition by cytotoxic T-cells. These events are at the heart of adaptive immunity and are critical in the elimination of virally infected and cancerous cells[10,11]. Import of peptides into the ER lumen depends on TAP1 and TAP2, ATP-binding cassette transporters embedded in the ER membrane[12-14]. Another component of the PLC is Tapasin (TPSN)[15], which recruits MHC-I to the PLC by binding to TAP1/2 via its transmembrane segment[16-19], and to MHC-I via its large luminal domain[20]. A luminal loop of TPSN also performs an editing function, ensuring that a high-affinity peptide binds the highly polymorphic groove on MHC-I[21-23]. TPSN has reduced affinity for peptide-loaded MHC-I molecules allowing them to dissociate from the PLC and traffic to the cell surface[21]. The functions of TPSN, TAP1 and TAP2 at the PLC are assisted by the luminal chaperones ERp57 and Calreticulin[24,25]. While the assembly, MHC-I loading, and editing at the PLC have been studied in detail, quality control processes regulating PLC function have not been identified.

Here, using an unbiased quantitative proteomics screen in mouse astrocytes, we identify the core PLC subunit TPSN as a substrate of the RNF185/MBRL complex and uncover the molecular basis for its ERAD recognition. We show that the RNF185/MBRL complex ensures that TPSN functions exclusively in the peptide loading complex. Loss of this ERAD-mediated fail-safe mechanism in RNF185 or MBRL-deficient cells results in increased MHC-I surface levels in professional antigen-presenting cells. Our findings highlight how the exquisite substrate specificity of ERAD is harnessed to ensure accurate antigen presentation. Considering the importance of antigen presentation in diseases such as cancer, our results suggest that TPSN regulation by ERAD may be exploited therapeutically to modulate immune surveillance.

## Results

### Tapasin levels are increased in RNF185/MBRL-deficient cells

To gain insight into the physiological role of the RNF185/MBRL ERAD complex, we sought to identify its endogenous substrates in mouse astrocytes, a cell type where MBRL function appeared to be critical[8,9]. We reasoned that endogenous substrates of this ERAD complex would be present at higher steady-state levels in MBRL KO cells. Therefore, whole-cell quantitative proteomics was used to compare protein abundance in astrocytes derived from MBRL KO mice with their WT littermates (Fig. 1a). CYP51A1 and TMUB2, two previously identified substrates of the RNF185/MBRL complex[7], were among the most increased proteins in MBRL KO astrocytes confirming the suitability of this approach for the identification of novel substrates (Fig. 1b, Source Data 2). Another highly enriched protein in MBRL KO astrocytes was Tapasin (TPSN), a single-spanning ER membrane protein essential for loading antigenic peptides into MHC-I molecules[26,27]. Western blotting analysis confirmed these results (Fig. 1c). Consistent with the experiments in mouse astrocytes, deletion of MBRL in human induced

pluripotent stem cells (iPSCs) (Fig. 1d), HEK293 (Fig. 1e), THP-1 (Supplementary Fig. 1a), and U2OS (Supplementary Fig. 1b) cells also resulted in higher TPSN steady-state levels. Importantly, the deletion of the MBRL-binding partner RNF185 in these cells resulted in a similar increase in TPSN steady-state levels (Fig. 1d, e). Moreover, the effect was specific as the levels of other ER proteins, including the ER chaperone ERp57, which is also a TPSN-binding partner and critical for MHC-I presentation[28], were unchanged (Fig. 1d, e, Supplementary Fig. 1a, b). Deletion of the ubiquitin ligase RNF5, more than 70% identical to RNF185[29], but unable to assemble with MBRL[7], did not affect TPSN steady-state levels in HEK293, THP-1, and U2OS cells. Therefore, ablation of the RNF185/MBRL ERAD complex results in a specific increase of TPSN steady-state levels in various cell types.

### Tapasin is a substrate of the RNF185/MBRL ERAD complex

The observations described above suggested that TPSN is a substrate of the RNF185/MBRL ERAD complex. To investigate this possibility further, we generated a TPSN transgene fused to sfGFP and HA-tags (TPSN-GFP-HA) expressed from a tetracycline responsive promotor. TPSN KO cells display a severe defect in MHC-I cell surface expression[15,30]. Expression of TPSN-GFP-HA in TPSN KO HEK293 cells restored MHC-I surface levels indicating that the fusion protein is functional (Fig. 2a). In agreement with these results, TPSN-GFP-HA assembled with its partners at the peptide loading complex as detected by immunoprecipitation followed by mass spectrometry (Fig. 2b, Source Data 3). Interestingly, TPSN also showed robust association with many ERAD factors, including MBRL. The association of TPSN with the RNF185/MBRL ERAD complex was independently confirmed by western blotting (Fig. 2c). Together these data showed that transgenic TPSN behaves like the endogenous protein and is a suitable tool to investigate TPSN regulation.

The steady-state levels of TPSN-GFP-HA were increased upon acute inhibition of the p97 ATPase or the proteasome, indicating that TPSN-GFP-HA was being degraded by ERAD (Supplementary Fig. 2a, b). Deletion of RNF185 or MBRL also resulted in higher steady-state levels of TPSN-GFP-HA, while deletion of the ERAD ligases RNF5 or HRD1 had little or no effect (Fig. 3a and Supplementary Fig. 2c). Importantly, re-expression of WT RNF185 or MBRL in their respective knockout lines restored normal steady-state levels of TPSN-GFP-HA. In contrast, re-expression of catalytically inactive RNF185 failed to rescue TPSN levels, indicating that the ubiquitin ligase activity is essential in controlling TPSN levels (Supplementary Fig. 2d, e). Depletion of TMUB1/2 (Supplementary Fig. 2f–h), part of the RNF185/MBRL complex, and UBE2K, the conjugating enzyme working with this complex, also resulted in increased steady-state levels of TPSN-GFP-HA (Supplementary Fig. 2i).

To analyse TPSN turnover we performed chase experiments upon translation shut-off with cycloheximide. These experiments confirmed that, in parental cells, TPSN-GFP-HA was a short-lived protein with a half-life of approximately 4 h (Fig. 3b). Importantly, degradation of TPSN-GFP-HA was blocked in RNF185 and MBRL KO cells while ablation of RNF5 did not affect TPSN turnover (Fig. 3b).

Degradation of ERAD substrates requires their prior ubiquitination[1,4]. We tested whether RNF185/MBRL stimulated TPSN degradation by promoting its ubiquitination. Consistent with its short half-life, TPSN ubiquitin-conjugates were readily detected in parental cells. (Fig. 3c). In contrast, RNF185 or MBRL KO cells showed much lower levels of ubiquitinated TPSN, even if their overall levels were higher in the KO cells. This effect was specific because cells deficient in the ERAD ubiquitin ligases RNF5 or HRD1 displayed levels of ubiquitinated TPSN comparable to controls (Fig. 3c).

There are four lysine residues within the short cytosolic tail of TPSN. Lysines are the most common acceptor sites for ubiquitination during protein degradation. We tested the importance of these lysine residues in TPSN degradation by mutating them to alanine in the TPSN 4K-to-A tail. This mutant showed increased steady-state levels

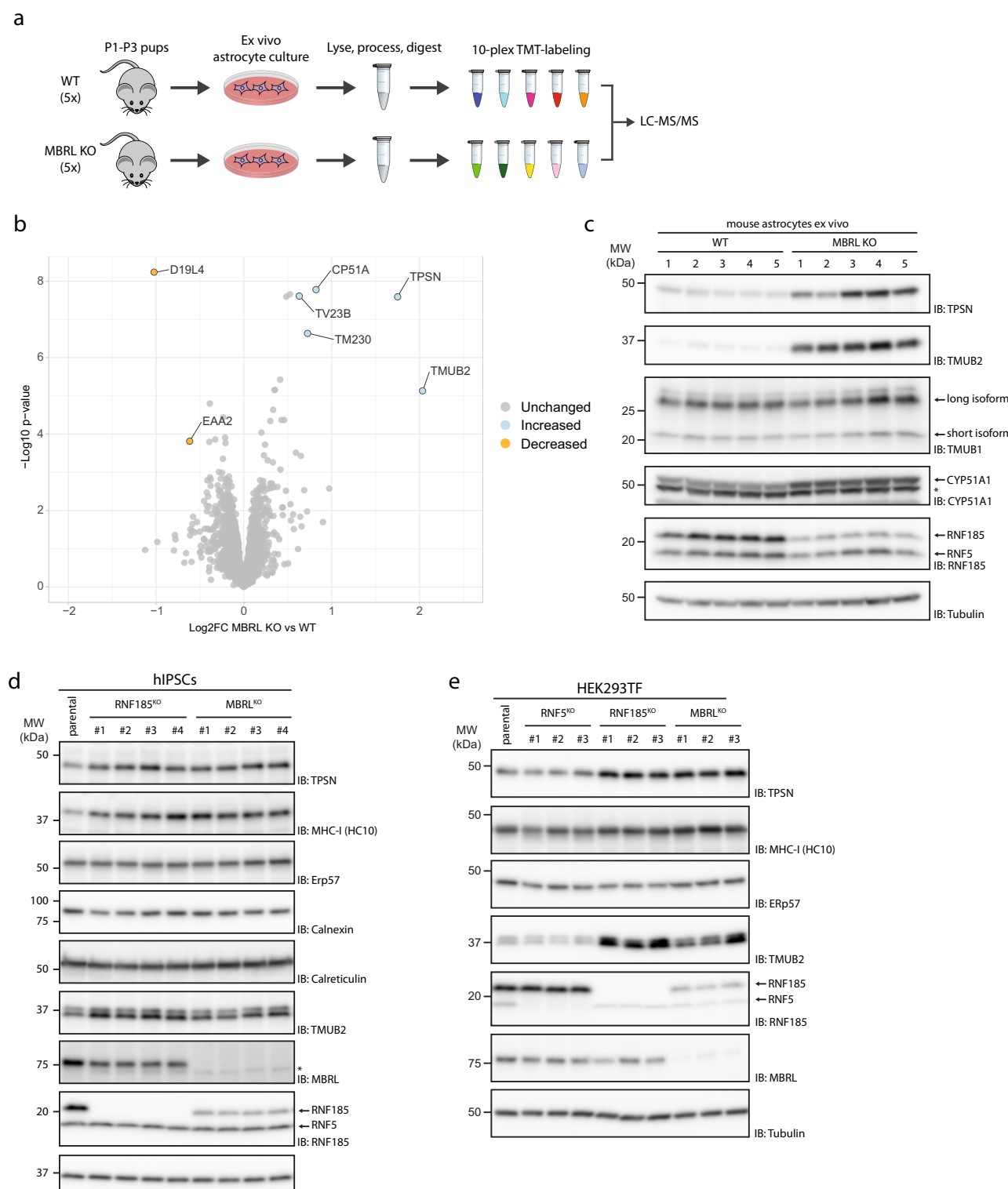

compared to WT TPSN and was largely resistant to an acute ERAD block upon p97 inhibition (Fig. 3d). Moreover, the TPSN 4K-to-A tail mutant was only poorly ubiquitinated (Fig. 3e), while its association with the RNF185/MBRL complex was unaffected (Supplementary Fig. 2j). Therefore, lysine residues in TPSN cytosolic tail are important for its ubiquitination but not for its binding to the RNF185/MBRL ERAD complex. Altogether, these data indicate that TPSN steady-state levels are regulated by RNF185/MBRL-mediated ERAD.

We previously observed that among the RNF185/MBRL complex components, MBRL appeared to be the one that co-precipitated most

efficiently with CYP51A1TM, another substrate of this ERAD complex[7]. This suggested that MBRL could play a role in selecting substrates for degradation. To test this possibility further, we asked whether MBRL could interact with TPSN independently of the other subunits of the RNF185/MBRL complex. Indeed, we observed that endogenous MBRL could precipitate TPSN in cells lacking RNF185 and TMUB1/2 (Fig. 3f). This association was specific as only background levels of TPSN were associated with the ERAD ligase HRD1. Therefore, MBRL binds TPSN independently of other components of the complex and likely plays an important role in TPSN recognition.

**Fig. 1 | RNF185/MBRL-deficient cells display elevated levels of the MHC-I chaperone TPSN. a** Schematic overview of the workflow for substrate identification in MBRL-deficient mouse astrocytes. Primary cortical astrocytes from 5 parental and 5 MBRL KO P1–P3 mouse pups were cultured ex vivo for 21 days. Astrocytes were harvested, lysed and protein extracts digested. The resulting peptide samples were labelled with TMT-10-plex reagent and analysed by LC-MS/MS. Moderated $t$-tests, with patient accounted for in the linear model, were performed using Limma, where proteins with $p$-value < 0.05 were considered as statistically significant. **b** Identification of proteins increased and decreased in MBRL KO mouse astrocytes compared to wildtype astrocytes. Volcano plot showing the relation of log2 fold-change and -log10 $p$-value of protein levels in MBRL KO vs. wildtype mouse astrocytes. Proteins that are significantly decreased in MBRL KO

astrocytes are on the left and labelled in Orange. Proteins that are significantly increased in MBRL KO astrocytes are on the right and labelled in Blue. Astrocytes from 5 animals were analysed for each condition. **c** Validation of the proteins identified in **b**. Protein extracts from wildtype and MBRL KO mouse astrocytes were analysed by SDS–PAGE and immunoblotting with the indicated antibodies. Of note, the anti-RNF185 antibody cross-reacts with RNF5, as indicated. The asterisk (*) indicates a non-specific background band. **d**, **e** TPSN protein levels are increased in RNF185 and MBRL KO human iPSCs and HEK293 cells. Extracts from parental, RNF185 and MBRL KO iPSCs (**d**) and HEK293 Flp-In T-Rex cells (**e**) were analysed by SDS–PAGE and immunoblotting with the indicated antibodies. The asterisk (*) indicates a non-specific background band.

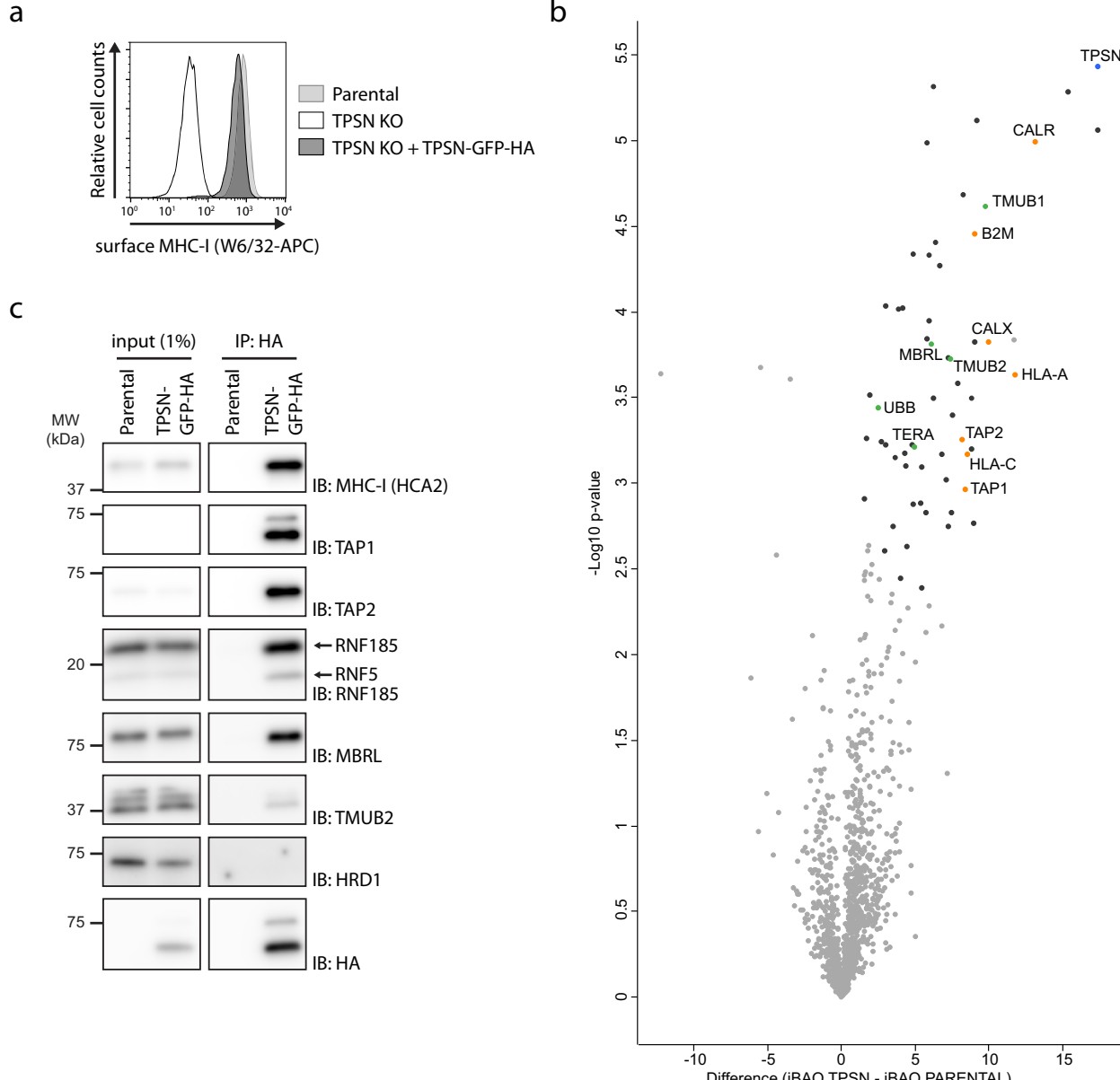

**Fig. 2 | TPSN-sfGFP-3xHA is functional and associates both with PLC and RNF185/MBRL ERAD complex. a** TPSN-sfGFP-3xHA is functional. Cell surface expression of MHC-I was analysed with the conformation-sensitive W6/32 antibody in Parental, TPSN KO, and TPSN KO HEK cells stably expressing TPSN-sfGFP-3xHA using flow cytometry. **b** Proteins co-precipitating with TPSN-sfGFP-3xHA were analysed by mass spectrometry. The $x$-axis shows enrichment of proteins associated with TPSN-sfGFP-3xHA over an untagged control. The $-\log_{10}$ of the $p$-value is

shown on the $y$-axis. $p$-values were calculated in Perseus using a two-sample $t$-test for significance. Proteins significantly enriched (FDR < 0.05) are displayed in black, orange (PLC components) or green (quality control factors). Three independent replicates were analysed. **c** Validation of the proteins co-precipitating with TPSN-sfGFP-3xHA from (**b**) as analysed by immunoblotting. Cells were lysed in buffer containing 1% DMNG and TPSN-sfGFP-3xHA was immunoprecipitated. Selected proteins were analysed by immunoblotting with the antibodies indicated.

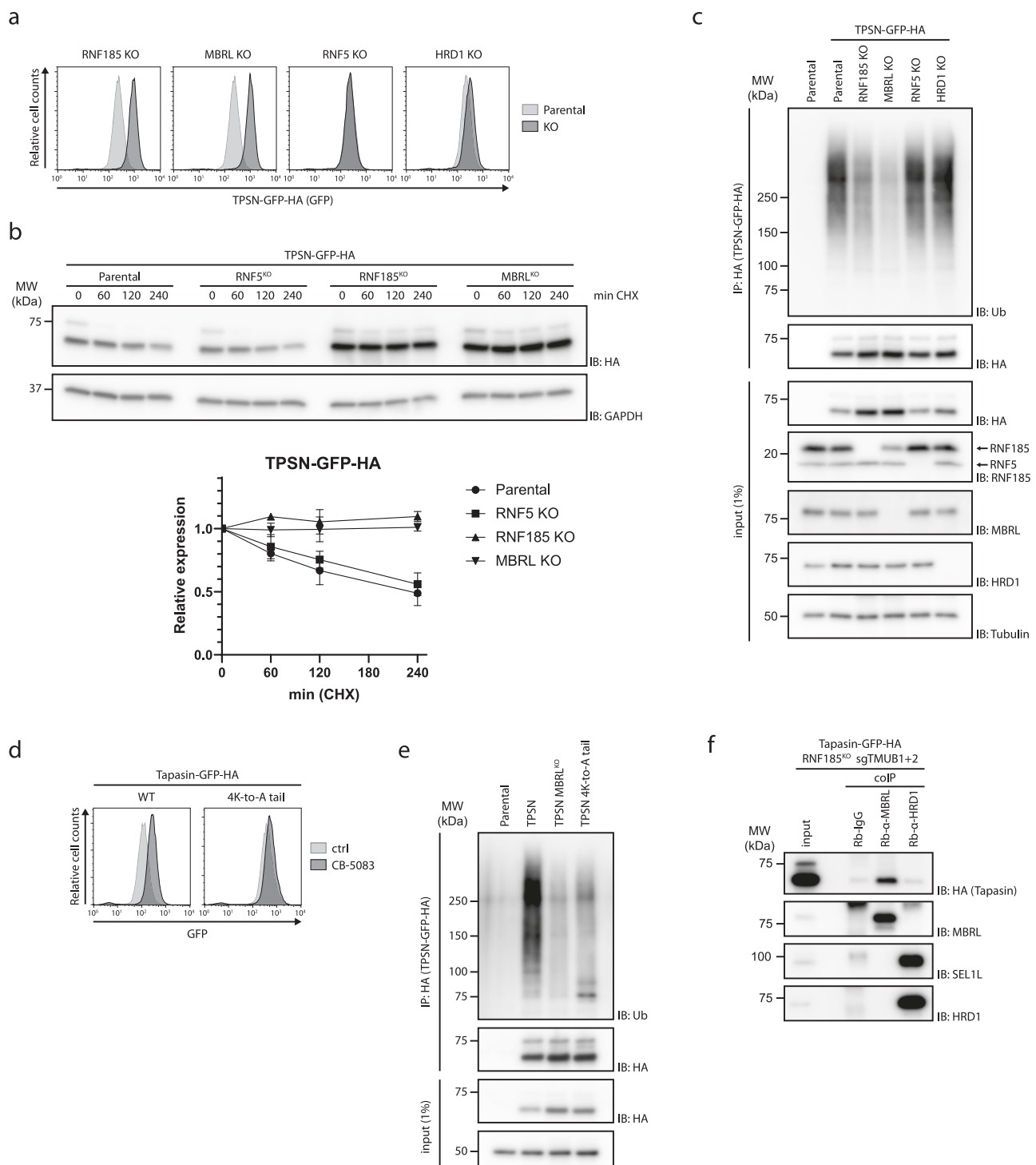

Previous studies showed that the poxvirus Molluscum contagiosum downregulates MHC-I surface expression of the host cells by promoting the rapid degradation of TPSN by an ERAD-like process[31]. This process requires the virally encoded protein MC80. We then tested whether MC80 was promoting TPSN degradation by hijacking the RNF185/MBRL ERAD complex. In agreement with previous studies[31], we observed that expression of MC80 resulted in lower TPSN steady-state levels that could be rescued upon inhibition of p97 or the proteasome (Supplementary Fig. 3a). However, deletion of RNF185 or MBRL did not interfere with MC80-induced degradation of TPSN (Supplementary Fig. 3b, c). Consistently, the ubiquitin-conjugating enzyme UBE2K, which assists RNF185/MBRL-dependent ubiquitination[7] was also dispensable for MC80-induced degradation of TPSN. In contrast, depletion of ubiquitin-conjugating UBE2G2 (Supplementary Fig. 3d) or the cognate ERAD ligases HRD1 and GP78 (Supplementary Fig. 3e) resulted in inhibition of TPSN degradation induced by MC80. Altogether, these data indicate that normally, TPSN degradation depends on the RNF185/MBRL complex, while in cells expressing the Molluscum contagiosum protein MC80, TPSN is degraded by distinct ERAD complexes.

## Charged residue in the membrane domain determines Tapasin assembly and degradation

Next, we investigated the conditions leading to TPSN degradation by the RNF185/MBRL complex. Our prior analysis did not identify any

**Fig. 3 | TPSN is a substrate of the RNF185/MBRL complex. a** TPSN levels are increased in RNF185 and MBRL knockout cells. Clonal RNF185, MBRL, RNF5, and HRD1 knockout lines were established from Flp-In T-Rex HEK293 cells expressing TPSN-sfGFP-3xHA. TPSN-sfGFP-3xHA levels were assessed by flow cytometry (based on GFP fluorescence). **b** RNF185/MBRL complex is essential for TPSN turn-over. Degradation of TPSN was analysed in parental, RNF5, RNF185, and MBRL KO Flp-In T-Rex HEK293 cells expressing TPSN-sfGFP-3xHA after inhibition of protein synthesis by cycloheximide (CHX). Cell extracts were analysed by SDS–PAGE and immunoblotting. TPSN was detected with an anti-HA antibody. GAPDH was used as a loading control and detected with an anti-GAPDH antibody. Quantifications of 3 independent experiments are shown in the graph below as mean; error bars represent the standard deviation. **c** RNF185/MBRL complex promotes TPSN ubi-quitination. TPSN-sfGFP-3xHA was immunoprecipitated from parental cells or cells lacking the indicated proteins and analysed by SDS–PAGE followed by immunoblotting with anti-HA and anti-ubiquitin antibodies. Parental cells were used as a negative control. **d** The steady-state levels of TPSN 4K-to-A tail mutant are high and largely unresponsive to ERAD inhibition. Flp-In T-Rex HEK293 cells expressing GFP-HA-tagged WT or 4K-to-A tail mutant TPSN were analysed by flow cytometry (based on GFP fluorescence) in the absence or presence of the p97 inhibitor CB-5083 (2.5 μM). **e** RNF185/MBRL complex promotes the ubiquitination of lysines in the cytosolic tail of TPSN. WT or 4K-to-A tail mutant TPSN expressed as GFP-HA fusion proteins were immunoprecipitated from the indicated cell lines and analysed by SDS–PAGE followed by immunoblotting with anti-HA and anti-ubiquitin antibodies. Parental cells were used as a negative control. **f** MBRL binds TPSN independently of its partners RNF185, TMUB1/2. Endogenous MBRL or HRD1 were immunoprecipitated from the indicated cells and precipitated proteins were analysed by immunoblotting with anti-HA (for TPSN-sfGFP-3xHA), anti-HRD1 and anti-SEL1L antibodies.

other components of the PLC as interactors of the RNF185/MBRL complex[7]. Therefore, we raised the possibility that RNF185/MBRL was interacting with and promoting the degradation of TPSN that was not assembled with its PLC partners. To test this possibility, we analysed TPSN levels in cells lacking TAP1, a TPSN partner at the PLC that transports antigenic peptides into the ER lumen for loading into the MHC-I[15]. Consistent with earlier observations[32], steady-state levels of TPSN were reduced in TAP1-deficient cells. Strikingly, deletion of RNF185 or MBRL in TAP1 KO cells restored normal TPSN levels sug-gesting that unassembled TPSN was indeed a substrate of the RNF185/MBRL complex (Fig. 4a and Supplementary Fig. 4a). The effect was specific as simultaneous deletion of TAP1 and the ubiquitin ligase RNF5 had little or no effect on TSPN levels. One interpretation of these results is that TPSN assembly with PLC components and its degrada-tion by the RNF185/MBRL complex are competing events. Consistent with this idea, we observed that overexpression of TAP1 resulted in increased TPSN steady-state levels (Fig. 4b) and stability (Fig. 4c). Interestingly, the turnover of overexpressed TAP1 occurs indepen-dently of MBRL (Fig. 4c). Collectively, these data indicate that RNF185/ MBRL promotes the degradation of unassembled TPSN and that once assembled with other PLC components, TPSN is no more degraded by ERAD.

Next, we investigated how unassembled TPSN was recognized by the RNF185/MBRL complex. TPSN assembles with TAP1 and TAP2 via its transmembrane segment[16–19]. Specifically, conserved positively charged lysine and negatively charged aspartate residues in TPSN and TAP1/2, respectively, form an intramembrane salt bridge important for the assembly of these proteins in the PLC[16] (Fig. 4d). We then asked whether these assembly determinants of TPSN were also important for its degradation by ERAD. To directly test this possibility, we generated soluble TPSN (sTPSN), a TPSN mutant lacking the C-terminal portion including the transmembrane segment[27], and TPSN K428A, with a single point mutation in the conserved intramembrane lysine residue at position 428. These proteins were expressed at high levels, and importantly, their steady-state levels were insensitive to p97 inhibition or deletion of the RNF185/MBRL complex, indicating that these TPSN mutants were not degraded by ERAD (Supplementary Fig. 4b, c). Indeed, cycloheximide chase experiments confirmed that in contrast to wt TPSN, TPSN K428A is a stable protein (Fig. 4e). Moreover, TPSN K428A failed to interact with the RNF185/MBRL complex consistent with it not being an ERAD substrate (Fig. 4f). Thus, the conserved intramembrane lysine at position 428 functions both as critical determinant of TPSN assembly into the PLC and as a degradation signal for the RNF185/MBRL ERAD complex. The dual role of the lysine 428 in assembly and degradation ensures that TPSN is functional only within the PLC, with molecules failing to assemble being quickly degraded in an RNF185/MBRL-dependent manner.

Seminal work on the assembly of the T-cell receptor (TCR) revealed the importance of intramembrane-charged residues in membrane protein quality control[33]. Specifically, the degradation of

unassembled TCR alpha chain (TCR-α) was shown to depend on the presence of two charged residues (lysine and arginine) within its transmembrane segment[34,35]. Given the parallels with our findings with TPSN, we asked whether RNF185/MBRL also played a role in the recognition of the charges within TCR-α that led to its ERAD. To this end, TCR-α transmembrane segment (TCR-α TM) was expressed as a fusion to the interleukin-2 receptor a chain (Tac antigen) as before (Supplementary Fig. 4d)[33] and its steady-state levels were assessed in various ERAD mutants (Supplementary Fig. 4e). Interestingly, steady-state levels of TCR-α TM were unaffected by RNF185/MBRL or RNF5 mutations but increased upon depletion of the ubiquitin ligase HRD1. These data suggest that while K428 is essential for TPSN recognition, RNF185/MBRL does not have a general role in the recognition of intramembrane charges during quality control. We also analysed CYP26A1 TM, a model ERAD substrate identified by our laboratory and that consists of the transmembrane segment of CYP26A1 (Sergejevs et al., in revision). Despite lacking intramembrane charges (Supplemen-tary Fig. 4d), CYP26A1 TM degradation requires RNF185/MBRL com-plex and is unaffected by depletion of RNF5 or HRD1 (Supplementary Fig. 4e). Therefore, intramembrane-charged residues may be essential for ERAD of some substrates, but they are not sufficient to determine specificity for an individual ERAD complex. Moreover, these data highlight the complexity of substrate recognition during membrane protein quality control.

## Increased MHC-I surface expression upon loss of RNF185/MBRL complex

The vital role of TPSN in MHC-I antigen presentation prompted us to test if its regulation by the RNF185/MBRL complex impacted the cell surface levels of MHC-I. To this end, we focused on macrophages, which are professional antigen-presenting cells, derived from human iPSCs. Parental, RNF185 and MBRL KO iPSC lines presented in Fig. 1d were differentiated into macrophages using a well-established protocol[36]. As observed in other cell types, loss of RNF185 or MBRL resulted in higher TPSN steady-state levels (Fig. 5a). Intriguingly, the increase in TPSN levels appeared more prominent in macrophages than in the corresponding iPSCs from where they were derived (compare Figs. 1d and 5a). The mutant macrophages also showed increased total levels of MHC-I as well as the TPSN partners at the PLC, TAP1 and TAP2. The increase was more pronounced for TAP1 than TAP2, in agreement with earlier observations showing the inter-dependence of the steady-state levels of TPSN, TAP1 and TAP2[18,32]. In contrast, the levels of several ER chaperones, including the TPSN-binding partner ERp57, were indistinguishable between parental and RNF185 or MBRL KO macrophages indicating that the effects observed were specific (Fig. 5a).

Upregulation of components of the antigen presentation pathway such as MHC-I and the PLC normally occurs upon activation of JAK-STAT signalling, either upon infection or stimulation of cells with the interferon gamma (IFNγ) cytokine[37]. Nevertheless, the increased levels

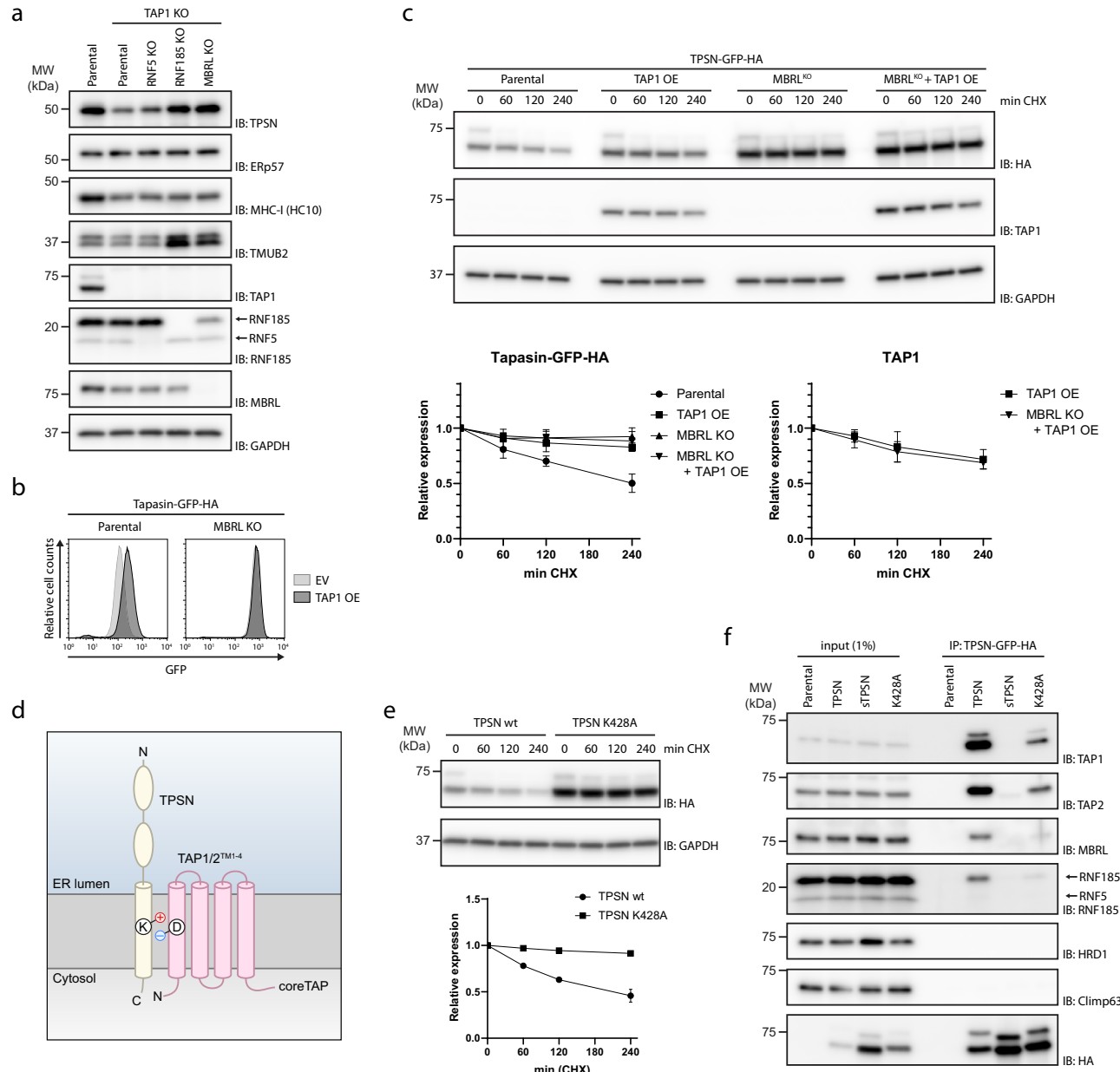

**Fig. 4 | TPSN assembly and degradation depends on evolutionary conserved charge in its transmembrane domain. a** Reduced TPSN steady-state levels in TAP1 depleted HEK293 cells depend on the RNF185/MBRL complex. RNF5, RNF185 and MBRL were deleted in TAP1 knockout cells. Cells were lysed, and protein extracts were subjected to SDS−PAGE followed by immunoblotting with the indicated antibodies. Parental cells were used as control. **b** TPSN steady-state levels are increased by overexpression of its binding partner TAP1. Parental and MBRL KO cells expressing TPSN-sfGFP-3xHA were transduced with a TAP1 overexpressing vector or an empty vector. TPSN-sfGFP-3xHA levels were assessed by flow cyto-metry (based on GFP fluorescence). **c** TPSN stability is increased by overexpression of its binding partner TAP1. Degradation of TPSN was analysed after inhibition of protein synthesis by cycloheximide (CHX) in Parental and MBRL KO cells expres-sing TPSN-sfGFP-3xHA in the presence and absence of a TAP1 overexpressing vector. Cell extracts were resolved by SDS−PAGE and subjected to immunoblotting for the proteins indicated. Quantifications for TPSN-sfGFP-3xHA and over-expressed TAP1 protein levels of 3 independent experiments are shown in the

graph below as mean; error bars represent the standard deviation. **d** Schematic representation of the interaction between TAP1/2 and TPSN mediated through an intramembrane salt bridge as described by Blees et al. (Reference #16). The lysine at position 428 of TPSN forms a salt bridge with an aspartic acid of TAP1 and TAP2, at position 32 and 16, respectively. **e** Degradation of TPSN and TPSN K428A was examined after inhibition of protein synthesis by cycloheximide (CHX). Cell extracts were analysed by SDS−PAGE and immunoblotting. TPSN and TPSN K428A were detected with anti-HA antibodies. GAPDH was used as a loading control and detected with an anti-GAPDH antibody. The graph shows the average of 3 experiments as the mean; error bars represent the standard deviation. **f** Immunoprecipitation of TPSN-sfGFP-3xHA and the indicated mutants. Cells expressing the indicated HA-tagged constructs were lysed in a buffer containing 1% DMNG, subjected to immunoprecipitation using anti-HA beads. Proteins were eluted and subjected to SDS−PAGE, followed by immunoblotting with the indicated antibodies.

of TPSN in untreated RNF185 or MBRL KO macrophages was not a consequence of constitutive activation of JAK-STAT signalling as monitored by the phosphorylation status of STAT1 tyrosine at position 701 (pSTAT1; Fig. 5a). Moreover, stimulation of the cells with IFNγ

resulted in upregulation of PLC components and MHC-I as well as STAT1 phosphorylation, indicating that JAK-STAT signalling is func-tional in RNF185 or MBRL KO macrophages. Importantly, RNF185 or MBRL KO macrophages showed higher steady-state levels of TPSN in

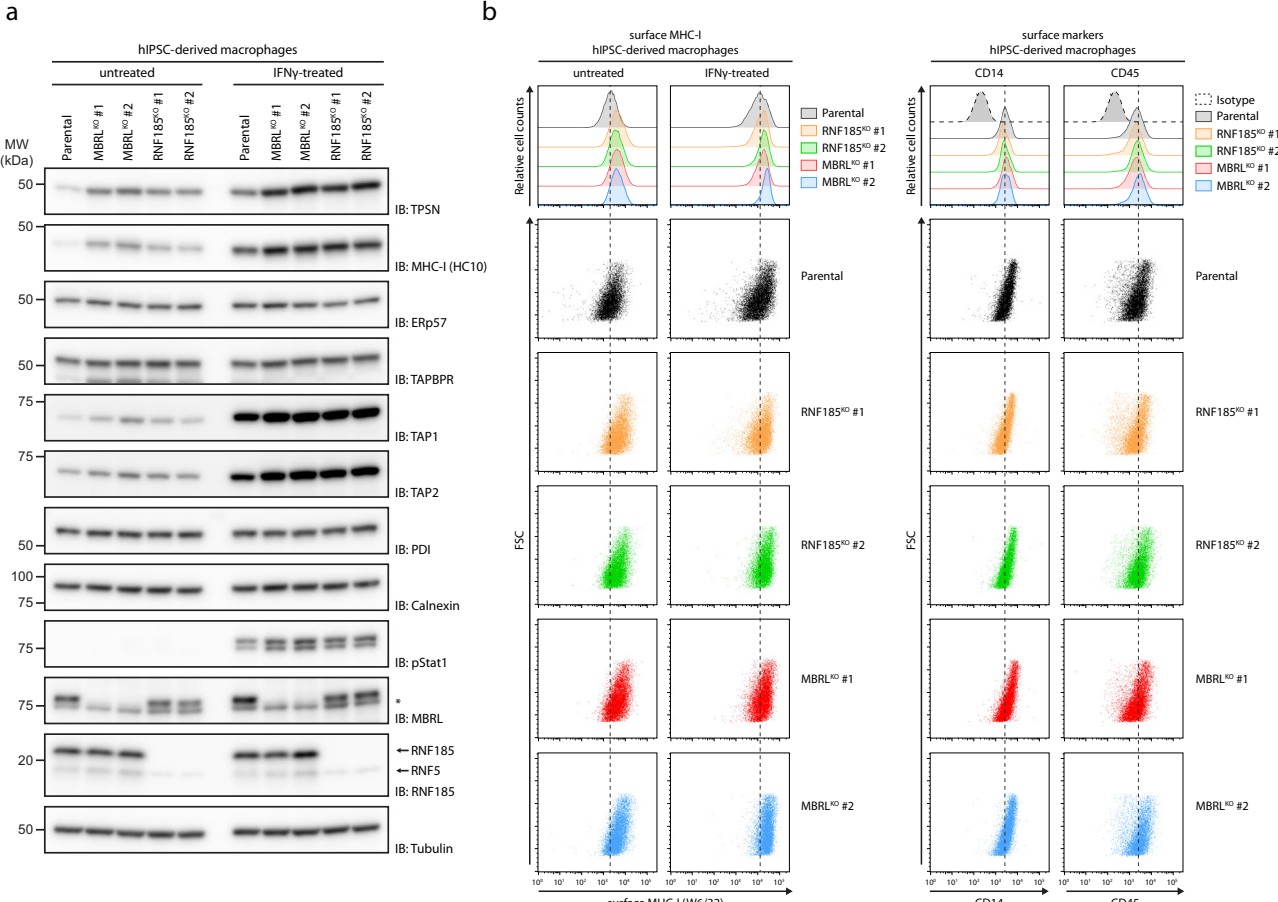

**Fig. 5 | Increased MHC-I surface expression in professional APCs upon loss of RNF185/MBRL ERAD complex. a** Deletion of RNF185 and MBRL in human iPSC-derived macrophages results in elevated TPSN levels. Parental, RNF185, and MBRL KO iPSCs from Fig. 1c were differentiated into macrophages. Macrophages were either left untreated or were treated for 16 h with IFN gamma (100 ng/mL). Protein extracts were analysed by SDS–PAGE and immunoblotting with the indicated antibodies. The asterisk (*) indicates a non-specific background band. **b** RNF185 and MBRL knockout iPSC-derived macrophages display elevated cell surface MHC-I levels as detected by flow cytometry with the conformation-sensitive W6/32 antibody. Macrophages were either left untreated or were treated for 16 h with IFN gamma (100 ng/mL). The identity of cells was confirmed with the macrophage markers CD14 and CD45.

relation to controls, even after IFNγ stimulation. Finally, we used a conformational sensitive antibody that detects peptide-loaded MHC-I molecules to assess the cell surface levels of MHC-I. Both under resting and IFN-stimulated conditions, loss of RNF185 or MBRL KO macrophages showed elevated surface levels of peptide-MHC-I complexes (Fig. 5b). The effects were specific for MHC-I, as the levels of cell surface proteins CD14 and CD45 were comparable between the RNF185 or MBRL KOs and control cells. Thus, the RNF185/MBRL ERAD complex regulates MHC-I surface expression.

## Discussion

The PLC is critical for the loading of antigenic peptides onto MHC-I molecules and adaptive immunity. While the process of peptide loading onto MHC-I by the PLC has been extensively studied, quality control mechanisms acting directly on PLC components are unknown. Here we identified an ERAD-based surveillance mechanism that regulates the levels of the core PLC component TPSN and that is important for MHC-I surface expression.

We showed that TPSN molecules that are not assembled with their PLC partners become substrates of the RNF185/MBRL ERAD complex. This process depends on the binding of RNF185/MBRL to the transmembrane segment of unassembled TPSN. Importantly, this binding involves an evolutionarily conserved lysine residue (K428) within the TPSN transmembrane domain that is also essential for its assembly in the PLC. In this case, K428 forms an intramembrane salt bridge with

conserved, negatively charged aspartate residues of its partners TAP1 or TAP2 (Fig. 6). Thus, both assembly and degradation of TPSN depend on K428. This mechanism is reminiscent of the one described for the assembly and quality control of the T-cell receptor (TCR)[33–35,38]. In this case, charged residues within the transmembrane segment of the TCR-α chain important for assembly with the CD3-δ subunit are also critical to trigger degradation of unassembled TCR-α[33,34]. While intramembrane-charged residues are common assembly determinants for a variety of protein complexes, how they are recognized by quality control factors is less clear. For example, RNF185/MBRL recognizes the intramembrane positively charged residue in TPSN, but not the ones in TCR-α. This highlights the exquisite specificity of the various ERAD complexes and suggests that additional determinants are involved in conferring their substrate specificity.

The higher TPSN levels observed upon loss of RNF185/MBRL manifested in increased surface expression of MHC-I in iPSC-derived macrophages. Concomitant with the rise in TPSN levels we observed an increase in the levels of TAP1 and TAP2, consistent with earlier observations[27]. The overall increase in these core PLC components may enhance peptide loading onto MHC-I, resulting in higher levels of peptide-MHC-I complexes at the cell surface as observed in human iPSC-derived macrophages deficient for RNF185/MBRL.

Besides recruiting MHC-I to the PLC, TPSN favours the loading of high-affinity peptides onto MHC-I through a process known as "peptide editing"[39,40]. The increase in TPSN levels likely reshapes the

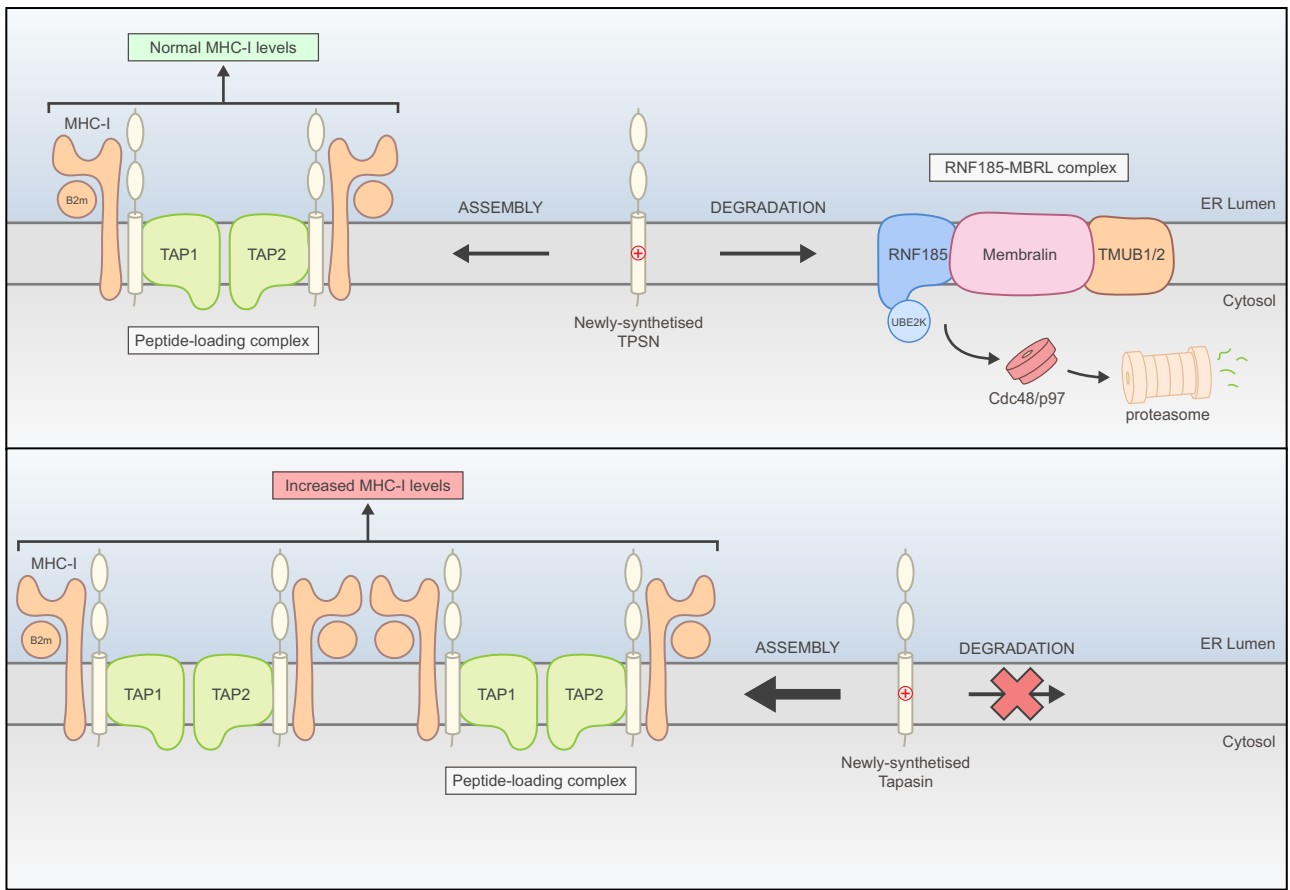

**Fig. 6 | Model for TPSN regulation by the RNF185/MBRL ERAD complex.** The RNF185/MBRL complex degrades a pool of unassembled TPSN, thereby limiting the amount available for assembly onto the PLC. RNF185/MBRL-dependent degradation involves the recognition of evolutionary conserved lysine residue in the TPSN transmembrane segment that is also essential for its assembly with TAP1 and TAP2. The absence of RNF185/MBRL results in elevated TPSN levels. These stabilize its partners TAP1 and TAP and ultimately result in increased levels of MHC-I. See text for details.

repertoire of antigenic peptides by favouring the loading of MHC-I with high-affinity peptides, which are more stable at the cell surface[41,42]. Increased MHC-I surface expression in RNF185/MBRL-deficient macrophages may also result from reduced turnover. These two possibilities are not mutually exclusive and future studies should clarify their individual contribution to the higher MHC-I surface expression observed in RNF185 and MBRL KO macrophages.

Both the levels and the repertoire of antigenic peptides presented by MHC-I are critical in eliciting cytotoxic T-cell responses[43–46]. Therefore, future studies should also assess if the changes in antigen presentation upon loss of RNF185/MBRL result in abnormal recognition by cytotoxic T cells and are eventually exploited therapeutically to modulate immune responses.

Earlier genetic studies revealed that an astrocyte-specific MBRL KO mouse developed a neuroinflammatory-like phenotype that resulted in motor neuron death[9]. In vitro, studies attributed this phenotype to the accumulation of extracellular glutamate through reducing the glutamate transporter EAAT2. While our proteomics analysis confirmed the reduction in EAAT2 levels, we wonder whether the dysregulated antigen presentation due to increased TPSN levels also contributes to the MBRL KO mouse phenotype. An increase in MHC-I presentation has been linked to neuroinflammation in different settings[47,48]. Therefore, future studies should test this possibility directly.

## Methods

Ethical approval for the mouse experiment was provided by the ethical committee at the Sanford Burnham Prebys Medical Discovery Institute.

### Cell lines

U2OS and THP-1 cells were obtained from the ECACC. The Lenti-X 293T cell line for the production of lentivirus was obtained from TakaraBio. Flp-In T-REx HEK293 cells were obtained from Invitrogen (Thermo Fischer Scientific). Flp-In T-REx HEK293 lines were established either as a clonal line or a polyclonal population according to the manufacturer's guidelines. Cells were grown at 37 °C 5% $CO_2$ in DMEM medium (Merck Life Science UK Limited #D6429) supplemented with L-Glutamine (2 mM; Gibco #25030024), Penicillin-Streptomycin (10 Units/mL; Gibco #15140122) and 10% FCS (Merck Life Science UK Limited #F9665). THP-1 cells were grown in RPMI medium (Merck Life Science UK Limited #R8758) supplemented with L-Glutamine, Penicillin-Streptomycin, and 10% FCS.

### Plasmids

The pcDNA5-FRT-TO and pOG44 plasmids were obtained from Invitrogen. cDNAs or sgRNAs, respectively, for protein overexpression and gene deletions, were cloned in a dual promoter lentiviral vector, as described previously[49]. The lentiviral TAP1 expression plasmid was kindly gifted by Emmanuel Wiertz.

### Lentivirus production

For gene transductions using lentiviruses, the virus was produced using Lenti-X 293T cells in 24-well plates using TransIT LT1 (Mirus Bio LLC #MIR 2305) and second-generation packaging vectors pMD2.G and psPAX2 according to standard lentiviral production protocols.

## Generation of CRISPR/Cas9-mediated knockout cells

For CRISPR/Cas9-mediated knockouts, cell lines were transfected using Mirus LT-1 according to the manufacturer's protocol. On the next day, cells were selected using Puromycin (2 μg/mL; Gibco #A1113803). After 48 h of selection, the selection medium was replaced with complete medium. To generate KO clones, cells were single-cell sorted using a BD FACSAria3 or AriaFusion. The knockout status of the clones was confirmed via flow cytometry and immunoblotting. To generate knockouts in THP-1 cells, cells were transduced in the presence of polybrene (Santa Cruz Biotechnology #28728-55-4) with lentivirus carrying CRISPR/Cas9. Two days after transduction, cells were put on 2 μg/mL Puromycin selection for 7 days. SgRNA sequences used are listed in Supplementary Table 1.

## Mice

All the mice were maintained in group housing on a 12-h light-dark cycle at 21 +/− 1 degree Celsius with ~50% humidity. Animals were given ad libitum access to food and water. The membralin[fl/fl] line was generated by inserting loxP sites in the introns preceding exon 2 and following exon 4 by Dr. Dongxian Zhang at Sanford Burnham Prebys Medical Discovery Institute (SBP) (The Jackson Laboratory #016574). Membralin homozygous KO animals were generated by crossing membralin[fl/fl] animals with Tg (*ACTB-Cre*) (The Jackson Laboratory #003376) to achieve germ-line deletion of membralin exons 2 to 4. Membralin heterozygous KO animals were crossed to maintain the colony and generate homozygous KO animals (mem-KO). The sex of the animals was not selected.

## Primary astrocyte culture

Primary cortical astrocytes were prepared from P1–P3 pups. Cortical tissue was treated with papain (80 U/mL; Worthington, cat# LS003126) for 20 min at 37 °C in an incubator and centrifuged for 5 min at 300 × *g*. Tissue pellets were resuspended in DMEM/F12 media supplemented with 10% Fetal Bovine Serum (FBS) (Nacalai USA, cat# 174012-500ML-BULK) and 1% penicillin/streptomycin (Thermo-Fisher Scientific, cat# 15140122) and mechanically dissociated using 1000 μL tips. Cells were seeded and cultured on 10 cm tissue culture-treated plates pre-coated with Matrigel (Corning, cat# 356237) for 21 days. The medium was changed every other day. Prior to harvesting the astrocyte protein samples, microglia were removed by shaking the plates at 100 rpm overnight in the incubator. The astrocytes were dissociated by 2.5% Trypsin (Thermo-Fisher Scientific, cat# 15090046) and washed in PBS for 5 times before being stored in −80 °C freezer.

## Proteome sample preparation from ex vivo mouse astrocyte culture

Mouse astrocytes were lysed in 5% SDS 50mM TEAB lysis buffer (pH 7.55). A total of 50 μg protein was reduced by incubation with 5 mM tris(2-carboxyethyl)phosphine (TCEP) for 15 min at 37 °C, and subsequently alkylated with 20 mM iodoacetamide for 30 min at room temperature in the dark. Protein digestion was performed using the suspension trapping (S-Trap™) sample preparation method using the manufacturer's guidelines (ProtiFi™, Huntington NY). Briefly, 2.5 μL of 12% phosphoric acid was added to each sample, followed by the addition of 165 μL S-Trap binding buffer (90% methanol in 100 mM TEAB, pH 7.1). This was added to the S-Trap Micro spin column. The samples were centrifuged at 4000 × *g* for 2 min until all the solution passed through the filter. Each S-Trap Mini-spin column was washed with 150 μL S-trap binding buffer by centrifugation at 4000 × *g* for 1 min. This process was repeated for a total of four washes. 25 μL of 50 mM TEAB, pH 8.0, containing trypsin (1:10 ratio of trypsin to protein) was added to each sample, followed by proteolytic digestion for 3 h at 47 °C using a thermomixer (Eppendorf) without shaking. Peptides were eluted with 50 mM TEAB pH 8.0 and centrifugation at 4000 × *g* for 2 min. Elution steps were repeated using 0.2% formic acid

and 0.2% formic acid in 50% acetonitrile, respectively. The three eluates from each sample were combined and dried using a speed-vac before storage at −80 °C.

## TMT-10 plex labelling

Each 15 μg protein digest was resuspended in 25 μL 100 mM HEPES, pH 8.5. TMT-10 plex labelling (TMT lot number: UG287488) was carried out as per the manufacturer's instructions. Samples were assigned to a TMT tag and 10 μL of the corresponding TMT tag was added per sample. Samples were incubated for 1 hr at room temperature. An aliquot corresponding to 0.5 μg was taken from each sample and pooled together for ratio and labelling efficiency checks, prior to making the full pooled sample. The test pool was quenched with 0.69 μL of 5% hydroxylamine, incubated for 15 min at room temperature, and dried using a speed-vac. The dried samples were cleaned up using C18 spin column as per the manufacturer's guidelines (Thermo Scientific), and subsequently dried using a speed-vac. Peptides were dissolved in 2% acetonitrile with 0.1% TFA, and the pooled sample was analysed for labelling efficiency and ratio check. For the ratio check, each sample (corresponding to a single TMT channel) was normalised to the median of all samples within its pool. Each sample was quenched with 4 μL 5% hydroxylamine and incubated for 15 min, and subsequently, samples were pooled together based on the scaling factors, which were calculated using the test pool. Samples were dried using a speed vac, cleaned using Pierce™ Peptide Desalting Spin columns as per the manufacturer's guidelines (Thermo Scientific), and dried down again using a speed vac prior to offline high-performance liquid chromatography (HPLC) fractionation.

## Offline HPLC fractionation

Peptides were resuspended in 55 μL ammonium formate, pH 8.0. Peptides were fractionated on a Basic Reverse Phase column (Gemini C18, 3 μm particle size, 110A pore, 3 mm internal diameter, 250 mm length, Phenomenex #00G-4439-Y0) on a Dionex Ultimate 3000 offline LC system. All solvents used were HPLC grade (Rathburn). Peptides (40 μL) were loaded on the column for 1 min at 250 μL/min using 99% Buffer A (20 mM ammonium formate, pH = 8) and eluted for 40 min on a linear gradient from 1 to 90 % Buffer B (100% ACN). Peptide elution was monitored by UV detection at 214 nm. Fractions were collected every 60 s from 2 min to 38 min for a total of 36 fractions. Fractions were pooled using non-consecutive concatenation to obtain 18 pooled fractions (e.g. pooled fraction 1: fraction 1 + 19). Each fraction was acidified to a final concentration of 1% TFA and dried using a speed vac.

## Mass spectrometry for TMT-10 plex samples

Peptides were dissolved in 2% acetonitrile with 0.1% TFA, and each sample was independently analysed on an Orbitrap Fusion Lumos Tribrid mass spectrometer (Thermo-Fisher Scientific), connected to a UltiMate 3000 RSLCnano System (Thermo-Fisher Scientific). Peptides (~2 μg per fraction) were injected on an Acclaim PepMap 100 C18 LC trap column (100 μm ID × 20 mm, 3 μm, 100 Å) followed by separation on an EASY-Spray nanoLC C18 column (75 μm ID × 50 cm, 2 μm, 100 Å) at a flow rate of 250 nl min⁻¹. Solvent A was water containing 0.1% formic acid, and solvent B was 80% acetonitrile containing 0.1% formic acid. The gradient used for the analysis of proteome samples was as follows: solvent B was maintained at 3% for 5 min, followed by an increase from 3 to 35% B in 120 min, 35–90% B in 0.5 min, maintained at 90% B for 4 min, followed by a decrease to 3% in 0.5 min and equilibration at 3% for 20 min.

Mass spectrometric identification and quantification were performed on an Orbitrap Fusion Tribrid mass spectrometer (Thermo-Fisher Scientific) operated in data-dependent, positive-ion mode. Full scan spectra were acquired in a range from 375 m/z to 1500 m/z, at a resolution of 120,000, with a standard automated gain control (AGC)

(Tune 3.3) and a maximum injection time of 50 ms. Precursor ions were isolated with a quadrupole mass filter width of 0.7 m/z and CID fragmentation was performed in one-step collision energy of 35% and 0.25 activation Q. Detection of MS/MS fragments was acquired in the linear ion trap in a rapid mode using a Top 3s method, with a standard AGC target and a maximum injection time of 50 ms. The dynamic exclusion of previously acquired precursor was enabled for 60 s with a tolerance of +/-10 ppm. Quantitative analysis of TMT-tagged peptides was performed using FTMS3 acquisition in the Orbitrap mass analyser operated at 60,000 resolution, with a standard AGC target and maximum injection time of 105 ms. HCD fragmentation on MS/MS fragments was performed in a one-step collision energy of 65% to ensure maximal TMT reporter ion yield and synchronous-precursor-selection (SPS) was enabled to include 10 MS/MS fragment ions in the FTMS3 scan.

All spectra were analysed using MaxQuant 1.6.10.43 and searched against SwissProt *mus musculus* fasta files (containing 25,350 database entries with isoforms, downloaded on 2021/03/10). Peak list generation was performed within MaxQuant and searches were performed using default parameters and the built-in Andromeda search engine. Reporter ion MS3 was used for quantification and the additional parameter of quantitation labels with 10 plex TMT on N-terminal or lysine was included. The enzyme specificity was set to consider fully tryptic peptides, and two missed cleavages were allowed. Oxidation of methionine and N-terminal acetylation were allowed as variable modifications. Carbamidomethylation of cysteine was allowed as a fixed modification. A protein and peptide false discovery rate (FDR) of less than 1% was employed in MaxQuant.

Reporter ion intensities were used for data analysis. Briefly, the data was filtered to remove proteins that matched to a contaminant or a reverse database, which were only identified by site, which were not quantified in every sample, or which contained less than 2 unique peptides. Reporter ion intensity values were log2 transformed. Moderated $t$-tests, with patients accounted for in the linear model, were performed using Limma, where proteins with $p$-value < 0.05 were considered as statistically significant. All analysis was performed using R 3.6.2.

### Co-immunoprecipitation
Cells were lysed in 1% DMNG (Anatrace #NG322) lysis buffer (50 mM Tris-HCl pH 7.5, 150 mM NaCl) containing a complete EDTA-free protease inhibitor cocktail (Roche #5056489001). Lysates were rotated for 60 min at 4 °C. Cell debris and nuclei were pelleted at 13,000 × g for 20 min at 4 °C. Post-nuclear supernatants were incubated for 2 h with anti-HA magnetic beads (Pierce, Thermo-Fisher Scientific). After four washes in 0.1% DMNG washing buffer, proteins were eluted in 1x sample buffer for 30 min at 37 °C. The eluate was transferred to a new tube and subsequently reduced using DTT (Merck Life Science UK Limited #D9779). Immunoblotting was performed as described below.

### Proteome sample preparation for Tapasin interactors
Immunoprecipitated samples on beads were resuspended in 25 μL 1x SDS sample buffer (5% SDS, 50 mM TAEB, pH 7.55). Disulfide bonds were reduced using 20 mM TCEP for 15 min at 47 °C. Following this, cooled samples were alkylated using 20 mM CAA in the dark for 15 min, and a 10% volume of 12% phosphoric acid was added to acidify the samples. S-trap binding buffer (90% methanol in 100 mM TAEB, pH 7.5) was added to acidified, denatured samples to a final volume of 190 μL and the resulting solution was loaded onto S-Trap micro spin columns (Protify), with a maximum of 150 μL of sample per load. Loaded spin columns were centrifuged at 4000 × g for 1 min, and this step was repeated until the entire sample was loaded onto a spin column. S-Trap columns were washed 5x with S-trap binding buffer (90% methanol in 100 mM TAEB, pH 7.5), and the columns were moved onto 2 mL low protein binding Eppendorf tubes. To each S-trap column, 25 μL of

digestion solution (50 mM TAEB, pH 8.0), containing 2 μg of Trypsin/Lys-C mix (Promega V5071) was added and loosely capped columns were incubated for 3 h at 47 °C on a ThermoMixer. Peptides were eluted with 30 μL of 50 mM TAEB, followed by 30 μL of 0.2% formic acid and 40 μL of 50% acetonitrile in 0.2% formic acid. Peptides were dried for 4 h at 37 °C in a vacuum centrifuge, and samples were stored at −80 °C until further analysis.

### Mass spectrometry for Tapasin interactors analysis
Peptides were dissolved in 2% acetonitrile containing 0.1% trifluoroacetic acid, and each sample was independently analysed on an Orbitrap Fusion Lumos Tribrid mass spectrometer (Thermo-Fisher Scientific), connected to an UltiMate 3000 RSLCnano System (Thermo-Fisher Scientific). Peptides (1 μg) were injected on a PepMap 100 C18 LC trap column (300 μm ID × 5 mm, 5 μm, 100 Å) followed by separation on an EASY-Spray nanoLC C18 column (75 μm ID × 50 cm, 2 μm, 100 Å) at a flow rate of 250 nl min⁻¹. Solvent A was water containing 0.1% formic acid, and solvent B was 80% acetonitrile containing 0.1% formic acid. The gradient used for analysis of proteome samples was as follows: solvent B was maintained at 2% for 5 min, followed by an increase from 2 to 35% B in 120 min, 35–90% B in 0.5 min, maintained at 90% B for 4 min, followed by a decrease to 3% in 0.5 min and equilibration at 2% for 10 min. The Orbitrap Fusion Lumos was operated in positive-ion data-dependent mode. The precursor ion scan (full scan) was performed in the Orbitrap in the range of 400–1600 m/z with a resolution of 120,000 at 200 m/z, an automatic gain control (AGC) target of $4 \times 10^5$, and an ion injection time of 50 ms. MS/MS spectra were acquired in the linear ion trap (IT) using Rapid scan mode after high-energy collisional dissociation (HCD) fragmentation. An HCD collision energy of 30% was used, the AGC target was set to $1 \times 10^4$, and dynamic injection time mode was allowed. The number of MS/MS events between full scans was determined on-the-fly to maintain a 3 s fixed duty cycle. Dynamic exclusion of ions within a ± 10 ppm m/z window was implemented using a 35 s exclusion duration. An electrospray voltage of 2.0 kV and capillary temperature of 275 °C, with no sheath and auxiliary gas flow, was used.

Data was analysed in Perseus v1.6.14.0. For pairwise comparison of two conditions (control vs Tapasin), two-sided $t$-test was performed (s0 = 0.1, FDR = 0.05, number of randomisations = 250). Data was visualised using scatter plot.

### Substrate ubiquitination experiments
Cells at around 80–90% confluency in a 6-well were lysed in RIPA buffer (50 mM Tris-HCl pH 7.5, 150 mM NaCl, 1% Triton X-100, 0.5% Sodium Deoxycholate, 0.1% SDS) containing NEM (20 mM; Merck Life Science UK Limited #E3876) and cOmplete protease inhibitor cocktail (Roche #5056489001). Lysates were rotated for 60 min at 4 °C. Cell debris and nuclei were pelleted at 13,000 × g for 20 min at 4 °C. Post-nuclear supernatants were incubated for 2 h with anti-HA magnetic beads (Thermo-Fisher Scientific #88837). After four washes in RIPA buffer, proteins were eluted in the sample buffer. The eluate was transferred to a new tube and subsequently reduced using DTT. Immunoblotting was performed as described below.

### Translation shut-off experiments
Cells were seeded in wells pre-coated with poly-L-lysine (Merck Life Science UK Limited #P8920). The next day, cells were incubated with cycloheximide (100 mg/mL; Merck Life Science UK Limited #C7698) for the indicated time points, after which cells were directly lysed in 1x sample buffer containing Benzonase (Merck Life Science UK Limited #E1014), cOmplete protease inhibitor cocktail (Roche #5056489001), and DTT. Lysates were incubated for 30 min at 37 °C, after which proteins were separated by SDS–PAGE. Immunoblotting was performed as described below.

## Immunoblotting

Samples were incubated at 37 °C for 15 min, separated by SDS–PAGE (Bio-Rad), and proteins were transferred to PVDF membranes (Bio-Rad). Membranes were probed with antibodies against indicated proteins. All antibodies used are listed in Supplementary Table 1. Reactive bands were detected by ECL (Western Lightning ECL Pro, Perkin Elmer #NEL121001EA), and visualized using an Amersham Imager 600 (GE Healthcare Life Sciences). Data quantification was performed using Image Studio software (Li-Cor), and graphs were plotted in GraphPad Prism v9.5. Representative images of three independent experiments are shown.

## Flow cytometry

For surface staining of MHC-I, cells were detached using EDTA, resuspended in FACS buffer (2% FBS, 2 mM EDTA in PBS), and washed once. Cells were incubated with W6/32-APC (1:50; Biolegend #311410) for 1 h at 4 °C. Next, cells were washed twice in FACS buffer and directly analysed using a BD LSRFortessa X-20 flow cytometer. For fluorescence measurement of GFP, cells were trypsinized, resuspended in FACS buffer, and directly analysed on the BD X-20. FACS data was analysed using FlowJo v10.

## iPSC culture

Human iPSC line SFC840-03-03[50] (https://ebisc.org/STBCi026-A) and KO clones were cultured in OXE8 medium as described previously[36]. Briefly, the iPSCs were cultured on hESC-qualified Geltrex (Thermo-FisherScientific, Cat# A1413302) coated plates and passaged using 0.5 mM EDTA in PBS. SNP-QCed frozen stocks were used, and the passage number was kept to within 3 passages from the QCed stock. 10 µM Rho kinase inhibitor (ROCKi) Y-27632 (Abcam, Cat# ab120129) was added to the culture medium during the first 24 h after thaw. Cells were incubated at 37 °C, 5% CO$_2$.

## Generation of iPSC knockouts

iPSCs were treated with ROCKi for 1 h prior to nucleofection. iPSCs were harvested using EDTA, pelleted, and resuspended in Buffer R (Invitrogen) at a concentration of 2.2E7 cells/mL. Cells were mixed with Alt-R™ AsCas12a (Cpf1) Ultra (IDT) crRNA complexes and nucleofection enhancer (IDT). Nucleofection was carried out using a Neon MPK5000 electroporator (Invitrogen) with the following parameters: 'HiTrans' 1400v, 20ms width, 1 pulse. Cells were then transferred to prewarmed ROCKi-containing OXE8 medium and allowed to recover. To grow iPSC KO clones, mitotically inactivated CF1 MEFs (Millipore #PMEF-CFL) were seeded in MEF medium (Advanced DMEM (Gibco #12491), 10% FCS, GlutaMAX (Gibco #35050), 50 µM 2-Mercaptoethanol (Gibco #31350)) on 0.1% gelatin (Sigma G1393) coated plates. The next day, a limiting dilution of iPSC lines were seeded on top of the MEFs in hES medium (KnockOut DMEM (Gibco #10829), 20% KnockOut Serum replacement (Gibco #10828), Non-Essential Amino Acids (Gibco #11140), GlutaMAX (Gibco), 50 µM 2 Mercaptoethanol (Gibco), Pen/Strep (Gibco #15140-122), 5 ng/mL bFGF (R&D #234-FSE/CF)) supplemented with ROCKi. 50% of hES medium was changed daily until colonies were ready for manual picking. KO clones were selected by Western blotting (Fig. 1d). Additionally, SNP-karyotyping was performed on genomic DNA to verify genomic integrity, using Illumina array Infinium GSA-24v3-0.

## Differentiation of iPSCs into macrophages

iPSCs were differentiated into macrophages using a protocol described previously[36], and described here in brief. To form embryoid bodies (EBs), 4 million iPSCs were seeded into an AggreWell 800 plate (STEMCELL Technologies, no. 34815) in EB medium (containing BMP4, VEGF and SCF) and incubated for 4 days at 37 °C, 5% CO2, with daily feeding of 75% medium change with EB medium. After 4 days, EBs were lifted from the plate and passed over a 40 µm cell strainer (Corning) to remove dead cells, before washing into a tissue culture plate with differentiation medium (containing IL-3 and M-CSF). EBs were used to set up two "factories" in two T175 flasks with 20 mL differentiation medium. Factories were incubated at 37 °C, 5% CO$_2$ and fresh 10 mL differentiation medium added weekly until macrophage precursors (PreMac) started to be produced. PreMac cells (emerging into the supernatant) were collected weekly and a minimum of equal volumes of differentiation media to the volume removed were replaced in the factory. PreMac were passed through a 40 µm cell strainer to obtain a single-cell suspension and plated in appropriately sized tissue culture plates for further culturing in OXM macrophage medium (containing M-SCF) for a further 7 days, with a 50% medium change on day 4.

## Flow cytometry of iPSC-derived macrophages

Cells were washed with PBS and lifted by incubation with Accutase (Gibco, #A11105-01). Then cells were stained in FACS buffer (PBS supplemented with 1% FBS, 10 µg/mL human-IgG (Sigma, no. I8640-100MG), and 0.01% sodium azide) and compared with isotype controls with the same fluorophores from the same company. Antibodies used included CD45-FITC (Immunotools, Cat# 21270453), CD14-FITC (Immunotools, Cat# 21270143), isotype-FITC (Immunotools, Cat# 21335013), W6/32-APC (Biolegend, Cat# 311400) and isotype-APC (Biolegend, Cat# 400220). Fluorescence was measured using the BD LSRFortessa X-20 (BD Biosciences) and analysed using FlowJo version 10.

## Statistics and reproducibility

Western blot data was quantified using Image Studio software (Li-Cor) and graphs were plotted using Prism (GraphPad). Representative images of at least three independent experiments are shown. Error bars represent the standard deviation, the measure of centre represents the mean.

## Reporting summary

Further information on research design is available in the Nature Portfolio Reporting Summary linked to this article.

## Data availability

The raw proteomics dataset generated during this study is available at PRIDE as PXD048728. The western blotting data generated during this study are available at Mendeley Data (https://doi.org/10.17632/vfd47jgr8t). Source data are provided in this paper.

## Materials availability

All unique reagents generated in this study are available from the lead contact without restriction. Further information and requests for reagents may be directed to and will be fulfilled by the lead contact, Pedro Carvalho (pedro.carvalho@path.ox.ac.uk).

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

## Acknowledgements

We thank P. Cresswell and L. Boyle for antibodies to TPSN and TAPBPR1, respectively, E. Wiertz for the lentiviral TAP1 expression plasmid, T. Rapoport and T. Williams for reading the manuscript, and O. Dushek, T. Elliot and P. Klenerman for discussions. P.C. was supported by an investigator award from the Wellcome Trust (223153/Z/21/Z). P.C. and M.L.v.d.W. were supported by a Research Grant from the BBSRC (BB/W001519/1). L.J. was supported by K99 AG066960 from the National Institute of Aging. S.A.C. was supported by the James Martin 21st Century Research Foundation. This research was co-funded by grant awards to M.T. (Wellcome Trust Multi-User Equipment grant (212947/Z/18/Z) and Investigator Award (215542/Z/19/Z)). R.J.K. and T.Y.H. received support from R01 AG062190 (NIA, NIH).

## Author contributions

M.L.v.d.W. and P.C. designed the study. M.L.v.d.W. performed most of the experiments with the help of N.S. L.J., T.Y.H. and R.J.K. provided the mouse astrocytes. T.H. prepared the TMT-10-plex samples from mouse astrocytes. M.E.D. and M.T. generated and analysed the mass spectrometry data. S.S. helped in the analysis of endogenous Tapasin and MHC-I. All work in human iPSCs was supervised S.A.C. and performed by K.S and M.L.v.d.W. M.L.v.d.W. and P.C. analysed the data with the help of all the authors. M.L.v.d.W. and P.C. wrote the manuscript with input from all the authors.

## Competing interests

The authors declare no competing interests.
