## [Peer Review File · Nature Communications]

Tapasin assembly surveillance by the RNF185/Membralin ubiquitin ligase complex regulates MHC-I surface expressionREVIEWER COMMENTS

Reviewer #1 (Remarks to the Author):

Review of the manuscript by van de Weijer et al. (Nat. Commun.):

In this manuscript van de Weijer and colleagues investigate the regulation of tapasin expression by the ubiquitin ligase complex RNF185/MBRL. The regulatory complex is part of the ERAD-mediated protein degradation machinery, eventually detecting misfolded proteins and marking them for degradation by the proteasome. Tapasin is a central component of the peptide loading complex that chaperones MHC class I molecules and also acts as a peptide exchange catalyst. Thus, antigen presentation and subsequent T cell activation events are conceivably modulated by the ubiquitination/degradation of tapasin by RNF185/MBRL.

The findings in this manuscript represent an interesting link between the ERAD components RNF185/MBRL and surface expression of MHC class I molecules. However, there are several major concerns regarding the manuscript.

(Authors: please provide page numbers (or better line numbers) with your manuscript!)

Page seven the authors write:

However, expression of TPSN-GFP-HA in TPSN KO HEK293 cells restored MHC-I antigen presentation indicating that the fusion protein is functional (Figure 2A).

While this experiment successfully restores MHC-I expression, it's important to note that the potential impact on peptide binding or the peptide repertoire remains undetermined. So instead of "restored MHC-I antigen presentation.." it should say "..restored MHC-I surface levels, indicating that.."

If the antibody detecting RNF185 also detects RNF5, as it looks in different IB (e.g. Frig. 3c), this should be noted in the paper.

Figure 3C – Ubiquitination of Tsn

An explanation of how this data was generated is not sufficient. It is not clear how this data shows that the ubiquitination of Tsn is lower than in the parental controls. While the UB-blot is shown for an GFP-TSN-IP, it could well be that the ubiquitination bands are originating from associated or contaminating proteins. The authors should perform mass spectrometry of the lanes in this gel to see what the most prominent proteins found to be modified are.

Regarding the role of Lys K428, many questions remain that have to be addressed. First, why is this lysine important for interaction with the RNF185/MBRL complex? It reads as if the authors assume a direct interaction. Would this not imply that a negatively charged residue in one of the RNF185/MBRL transmembrane helices interacts? Is there such a candidate Asp/Glu residue available. They could then mutate this residue and see whether it compromises the interaction with tapasin. In the discussion the authors refer to their own unpublished observations. These observations should be shared and complemented by the suggested mutational studies. Moreover, the authors should perform experiments that delineate the lysines that are ubiquitinated. Tapasin contains several lysines at the very C-terminus that are likely candidates. Upon mutation of this lysine cluster ubiquitination should be abolished and surface levels of MHC-I upregulated.

There is a paper by Daved Fremont's lab: "Molluscum contagiosum virus MC80 sabotages MHC-I antigen presentation by targeting tapasin for ER-associated degradation" (Plos Pathogens (2019) where they already suggest that the tapasin tail becomes ubiquitinated and invoked ERAD. I find it very surprising that the authors do not cite this publication. Were they not aware or did they leave out for a reason? Definitely, the findings of this paper should be discussed in the light of their own results.

Figure 4D – Loading control is missing and explanation is not sufficient.

Figure 5A - The MBRL antibody detects two bands in this IB, even though it detects only one in all other blots (besides Figure 1D and S1A). Next to the label "MBRL" is a star (*) that also appears in other Figures, but the meaning of this star is not explained.

Figure S2E should show that the catalytic activity of RNF185 is important for the degradation of Tsn, but somehow, the expression levels of both variants differ. WT seems to be expressed in smaller amounts. Why?

Last sentence results: Thus, RNF185/MBRL ERAD complex regulates MHC-I antigen presentation.

Sure? It seems to regulate the amount of Tsn, which influences the amount of MHC-I on the cell surface, but the paper did not analyze peptides bound to MHC-I; therefore, the authors except for

the discussion, should only make statements about the expression-level/cell-surface level of MHC-I, not antigen presentation. Otherwise they should show experimentally that a tapasin-dependent repertoire of peptides is bound.

Reviewer #2 (Remarks to the Author):

Reviewer #3 (Remarks to the Author):

The manuscript by Carvalho and colleagues dissects the function of an endoplasmic reticulum (ER)- associated ubiquitin ligase complex comprised of the RING-finger protein RNF185, Membralin, and a member of the TMUB family in quality control of the peptide loading complex (PLC) in the context of antigen presentation. The same group previously identified this protein complex as a quality control ubiquitin ligase, involved in ER-associated protein degradation (ERAD). Genetic knockout of Membralin, an essential subunit of this ligase complex, causes embryonic lethality in mice, but the underlying mechanism is unclear.

Using a quantitative mass spectrometry method, the authors searched for endogenous substrates of the Membralin complex and found that the steady-state level of Tapasin, a subunit of the PLC, is significantly increased in Membralin knockout astrocytes. They demonstrated that this is caused by reduced degradation. They further show that the Membralin ligase complex binds unassembled Tapasin, which is required for efficient ubiquitination and degradation of unassembled Tapasin. An intramembranously located charged residue is critical for this regulation, which appears to influence the surface presentation of MHC class I molecules.

The experimental designs are straightforward, and the data presented are clean and support the main conclusions. However, in its current form, the paper has a relatively narrow scope, and the physiological relevance of the reported findings needs to be further clarified. I hope that addressing some of the questions listed below will help the authors improve the paper, making it a strong candidate for Nat. Comm.

Main points

1. From the study, it is still unclear whether the Membranlin complex has a general role in the quality control of unassembled membrane proteins carrying exposed charged residues in their transmembrane domains and, if not, what determines the substrate specificity. The authors may address this question by presenting more details about their proteomic analyses. For example, among the proteins identified, how many are ER-synthesized membrane proteins? How many are components of protein complexes? Do many of them carry charged residues in their transmembrane domains? Another approach is to check previously published ERAD substrates known to have charged residues in the transmembrane domains. The best-characterized example is the TCRalpha chain, which carries two conserved lysines in its transmembrane domain. The authors mentioned it in the discussion and suggested that the degradation of TCRalpha is not dependent on the Membranlin complex. They should include this data and discuss possible models to explain how the Membranlin complex may select specific substrates but ignore others with a similar lesion.

2. Another central open question is how a quality control ligase recognizes substrates bearing a charged residue in a transmembrane domain. Fully addressing this question is undoubtedly beyond the scope of this study. However, the authors can include more experiments to bolster the concept that the Membranlin complex can recognize an unassembled membrane protein with a charged residue in the transmembrane domain. For example, does the ligase complex also degrade proteins bearing negatively charged residues in the membrane? Which subunit of the complex is responsible for substrate recognition? Is substrate recognition mediated by the transmembrane domains of a ligase subunit? Is there any unique conserved sequence feature in the ligase that may allow such recognition?

3. The authors show that in the absence of the Membrane complex, more unassembled Tapasin is present in the cell. Coincidentally, more MHC molecules are found on the cell surface. Are these two observations functionally linked? If so, why can more unassembled Tapasin help more MHC reach the cell surface? The experiment in Figure 5 suggests a possibility that contradicts the proposed quality control function of the Membranlin complex. This ligase seems to degrade Tapasin and TAP1/TAP2 after these proteins have assembled into a functional complex. Can the authors design an experiment to elucidate whether the Membranlin complex targets unassembled Tapasin during PLC biogenesis or mature PLC after the complex is disassembled (which may be an integral regulatory step in peptide loading)? If it is the latter case, it will deviate the study from the previously reported paradigm of membrane protein quality control and, thus, significantly improve the study's novelty.

4. Is the degradation defect of PLC relevant to the in vivo Membranlin knockout mouse phenotype? Do the Membranlin knockout astrocytes show signs of inflammation or astrogliosis that can be suppressed by knocking down the PLC. The authors may use qRT-PCR to check the expression of well-characterized cytokines and chemokines involved in astrocyte activation. Results along this line may explain the observed CNS dysfunction and animal death phenotypes.

Minor points:

The authors should include some subheadings to improve the readability of the result section.

Reviewer #4 (Remarks to the Author):

The manuscript by van de Weijer et al describes an ER associated degradative (ERAD) pathway that acts on tapasin to regulate levels of cell surface MHC-I. The three components of the pathway and a selection of substrates were previously identified by the authors in astrocytes and the present manuscript extends these earlier observations to tapasin. An unbiased proteomics screen on ERAD KO cells showed enrichment for tapasin implicating this degradative pathway in controlling steady state tapasin levels. The pathway is shown to function not only in astrocytes but also in a variety of conventional cell lines. Interestingly this ERAD pathway selectively targets only tapasin among the members of the peptide loading complex (PLC). The authors identify conserved a lysine in the transmembrane domain of tapasin as crucial for targeting by the ERAD pathway. The same lysine is also necessary for interaction with the TAP transporter for proper PLC functioning. Overall the experiments are well carried out, the data clearly presented, and the writing is concise. I have only a few minor points:

1. In the second para of the Results section: “restored MHC-I antigen presentation” is not strictly correct as no assay was performed for presentation, e.g. with a T cell read out. Instead “restored surface levels of MHC-I” would be the more appropriate phrasing.

2. Can the authors speculate on the efficiency of this ERAD pathway in different tissues? Since surface MHC-I levels are known to vary among different cell types, could these differences be ascribed partly to the upregulation or downregulation of components of the ERAD pathway.

3. In Methods under the subheading “Proteome sample preparation...”, the first line states “A total of 50 ug protein was reduced...”. However I could not find a description of how cell lysates were prepared for this step. Were the lysates prepared using 1% DMNG as described under “Co-immunoprecipitation”? Also, that subheading should be “from ex vivo mouse astrocyte culture” not “to ex vivo mouse astrocyte culture”.

Reviewer #1 (Remarks to the Author)

Review of the manuscript by van de Weijer et al. (Nat. Commun.):

In this manuscript van de Weijer and colleagues investigate the regulation of tapasin expression by the ubiquitin ligase complex RNF185/MBRL. The regulatory complex is part of the ERAD-mediated protein degradation machinery, eventually detecting misfolded proteins and marking them for degradation by the proteasome. Tapasin is a central component of the peptide loading complex that chaperones MHC class I molecules and also acts as a peptide exchange catalyst. Thus, antigen presentation and subsequent T cell activation events are conceivably modulated by the ubiquitination/degradation of tapasin by RNF185/MBRL.

The findings in this manuscript represent an interesting link between the ERAD components RNF185/MBRL and surface expression of MHC class I molecules. However, there are several major concerns regarding the manuscript.

(Authors: please provide page numbers (or better line numbers) with your manuscript!)

1- We are glad that the reviewer found our study interesting. We have now added page and line numbers, as suggested.

Page seven the authors write:

However, expression of TPSN-GFP-HA in TPSN KO HEK293 cells restored MHC-I antigen presentation indicating that the fusion protein is functional (Figure 2A).

While this experiment successfully restores MHC-I expression, it's important to note that the potential impact on peptide binding or the peptide repertoire remains undetermined. So instead of "restored MHC-I antigen presentation.." it should say "..restored MHC-I surface levels, indicating that.."

2- We agree with the reviewer and modified the text (on page 7 and beyond) to more accurately describe our findings.

If the antibody detecting RNF185 also detects RNF5, as it looks in different IB (e.g. Frig. 3c), this should be noted in the paper.

3- Our previous paper on the identification of the RNF185/MBRL ubiquitin ligase complex showed that the anti-RNF185 antibody also recognizes RNF5, albeit more weakly (PMID: 32738194). To avoid any potential confusion, we have now added a note in the legend of Figure 1c.

Figure 3C – Ubiquitination of Tsn

An explanation of how this data was generated is not sufficient. It is not clear how this data shows that the ubiquitination of Tsn is lower than in the parental controls. While the UB-blot is shown for an GFP-TSN-IP, it could well be that the ubiquitination bands are originating from associated or contaminating proteins. The authors should perform

mass spectrometry of the lanes in this gel to see what the most prominent proteins found to be modified are.

4- We apologize if the data on TPSN ubiquitination was not clearly described. To detect TPSN ubiquitination, we used a well-established protocol in the field and that is commonly used by our lab (for example PMID: 32738194 and PMID: 36318477). In this protocol, cells are lysed in a harsh buffer that disrupts most protein-protein interactions, ensuring that the precipitating ubiquitin molecules are covalently conjugated to the immunoprecipitated protein, in this case TPSN. A detailed description of the protocol is included in the Material & Methods (Substrate ubiquitination experiments section). We show that in RNF185 and MBRL KO cells while TPSN levels are increased, ubiquitinated TPSN levels are reduced in relation to parental cells or cells lacking other ERAD ubiquitin ligases such as RNF5 or HRD1. These data show that RNF185/MBRL specifically regulate TPSN ubiquitination. Moreover, we now show that TPSN 4K-to-A tail, a mutant in which the 4 lysine residues in the cytosolic TPSN tail were mutated to alanine, is poorly ubiquitinated (Figure 3d, 3e, and Supplementary Figure 2j). Like WT TPSN, this mutant assembles with PLC-components and interacts with RNF185/MBRL. Altogether, these experiments support that the TPSN is directly ubiquitinated in RNF185/MBRL-dependent manner and that the four lysine residues in the cytosolic tail of TPSN are the main acceptor sites for ubiquitin modification.

Regarding the role of Lys K428, many questions remain that have to be addressed. First, why is this lysine important for interaction with the RNF185/MBRL complex? It reads as if the authors assume a direct interaction. Would this not imply that a negatively charged residue in one of the RNF185/MBRL transmembrane helices interacts? Is there such a candidate Asp/Glu residue available. They could then mutate this residue and see whether it compromises the interaction with tapasin. In the discussion the authors refer to their own unpublished observations. These observations should be shared and complemented by the suggested mutational studies.

5- The mechanism of substrate recognition by MBRL is an important but non-trivial question to address. It is a topic of active research in our lab but, as pointed out by reviewer #2, falls outside of the scope of this study. Nevertheless, we now include additional data on the determinants recognized by RNF185/MBRL. We demonstrate that while the lysine at position 428 is essential for the recognition of TPSN, the presence of intramembrane charges is not a general feature of RNF185/MBRL substrates. This is illustrated by the case of CYP26A1TM, also a substrate of the RNF185/MBRL ERAD complex but that does not contain any charge within its TMD. On the other hand, the presence of charges is not sufficient for RNF185/MBRL recognition as illustrated by the case of the T-cell receptor alpha chain (TCR α), a well characterized ERAD substrate. Ground breaking work by Bonifacino and colleagues in the early 1990's showed that degradation of unassembled TCR α required two positively charged residues (arginine, lysine) within its TMD. We now show that TCR α TMD is degraded independently of RNF185/MBRL, requiring instead the Hrd1 ERAD complex. These new

data (Supplementary Figures 4d, 4e) illustrates both the specificity and the complexity of membrane substrate recognition during ERAD.

In our efforts to understand substrate recognition by RNF185/MBRL, we mutagenized K428 to a variety of residues. We observed that swapping the positively charged lysine to negatively charged aspartate (K428D) results in efficient recognition and degradation in a RNF185/MBRL-dependent manner. We thus favour a model in which RNF185/MBRL rather than recognizing the charges directly, detects specific behaviour of the TMD within the membrane (such as distortion within the bilayer for example due to snorkelling of the charged residues towards the polar phospholipid head groups). At the moment this is speculative and work in progress, therefore we do not include these data in the manuscript.

Moreover, the authors should perform experiments that delineate the lysines that are ubiquitinated. Tapasin contains several lysines at the very C-terminus that are likely candidates. Upon mutation of this lysine cluster ubiquitination should be abolished and surface levels of MHC-I upregulated.

6- We thank the reviewer for this suggestion. As suggested, we generated a mutant in which the 4 lysine residues in the cytosolic tail of TPSN were mutated to alanine (TPSN 4K-to-A tail). Like wild type TPSN, this mutant interacts both with PLC components and the RNF185/MBRL complex. However, TPSN 4K-to-A tail ubiquitination is strongly reduced indicating that the lysine residues in the cytosolic tail are the primary acceptor sites for RNF185/MBRL ubiquitination. Consistent with the TPSN 4K-to-A tail being poorly ubiquitinated, its levels are largely insensitive to ERAD inhibition (Figures 3d, 3e and Supplementary Figure 2j).

There is a paper by Daved Fremont's lab: "Molluscum contagiosum virus MC80 sabotages MHC-I antigen presentation by targeting tapasin for ER-associated degradation" (Plos Pathogens (2019) where they already suggest that the tapasin tail becomes ubiquitinated and invoked ERAD. I find it very surprising that the authors do not cite this publication. Were they not aware or did they leave out for a reason? Definitely, the findings of this paper should be discussed in the light of their own results.

7- We thank the reviewer for pointing this out. We now include data showing that MC80-triggered degradation of TPSN follows a distinct ERAD route that is independent of RNF185/MBRL (Supplementary Figure 3).

Figure 4D – Loading control is missing and explanation is not sufficient.

8- This point raised by the reviewer is not clear to us. In Figure 4d (now Figure 4f), the Input lanes show that equal amounts of lysate were used for in the immunoprecipitation, as confirmed by the comparable of levels of endogenous TAP1,

TAP2, RNF185, MBRL and HRD1. The results show that wild type TPSN interacts with PLC and RNF185/MBRL while those interactions are compromised in sTPSN and TPSN K428A. but not another ERAD component HRD1 or the abundant ER membrane protein CLIMP63. Importantly we show that wild type TPSN does not co-precipitate with the ERAD component HRD1 indicating that the interactions detected are specific. We are confident that the experiment is well controlled both for loading and specificity.

Figure 5A - The MBRL antibody detects two bands in this IB, even though it detects only one in all other blots (besides Figure 1D and S1A). Next to the label "MBRL" is a star (*) that also appears in other Figures, but the meaning of this star is not explained.

9- We thank the reviewer for pointing this out. We have now indicated this in the relevant figure legends. As for the explanation, in some cell lines, the MBRL antibody produces a non-specific background band slightly below the specific MBRL band.

Figure S2E should show that the catalytic activity of RNF185 is important for the degradation of Tsn, but somehow, the expression levels of both variants differ. WT seems to be expressed in smaller amounts. Why?

10- We have observed previously that the levels of the RNF185 catalytic mutant are slightly higher than the wildtype (PMID: 32738194). This is not uncommon for E3 ligases, as many can self-ubiquitinate resulting in their own degradation. Importantly, this does not alter the conclusion that RNF185 ubiquitin ligase activity is necessary for TPSN degradation.

Last sentence results: Thus, RNF185/MBRL ERAD complex regulates MHC-I antigen presentation.

Sure? It seems to regulate the amount of Tsn, which influences the amount of MHC-I on the cell surface, but the paper did not analyze peptides bound to MHC-I; therefore, the authors except for the discussion, should only make statements about the expression-level/cell-surface level of MHC-I, not antigen presentation. Otherwise they should show experimentally that a tapasin-dependent repertoire of peptides is bound.

11- We thank the reviewer for highlighting this. This ties together with the first comment, and we have amended this throughout the manuscript.

Reviewer #2 (Remarks to the Author)

Reviewer #3 (Remarks to the Author)

The manuscript by Carvalho and colleagues dissects the function of an endoplasmic reticulum (ER)- associated ubiquitin ligase complex comprised of the RING-finger protein RNF185, Membralin, and a member of the TMUB family in quality control of the peptide loading complex (PLC) in the context of antigen presentation. The same group previously identified this protein complex as a quality control ubiquitin ligase, involved in ER-associated protein degradation (ERAD). Genetic knockout of Membralin, an essential subunit of this ligase complex, causes embryonic lethality in mice, but the underlying mechanism is unclear.

Using a quantitative mass spectrometry method, the authors searched for endogenous substrates of the Membralin complex and found that the steady-state level of Tapasin, a subunit of the PLC, is significantly increased in Membralin knockout astrocytes. They demonstrated that this is caused by reduced degradation. They further show that the Membralin ligase complex binds unassembled Tapasin, which is required for efficient ubiquitination and degradation of unassembled Tapasin. An intramembranously located charged residue is critical for this regulation, which appears to influence the surface presentation of MHC class I molecules.

The experimental designs are straightforward, and the data presented are clean and support the main conclusions. However, in its current form, the paper has a relatively narrow scope, and the physiological relevance of the reported findings needs to be further clarified. I hope that addressing some of the questions listed below will help the authors improve the paper, making it a strong candidate for Nat. Comm.

Main points

1. From the study, it is still unclear whether the Membralin complex has a general role in the quality control of unassembled membrane proteins carrying exposed charged residues in their transmembrane domains and, if not, what determines the substrate specificity. The authors may address this question by presenting more details about their proteomic analyses. For example, among the proteins identified, how many are ER-synthesized membrane proteins? How many are components of protein complexes? Do many of them carry charged residues in their transmembrane domains? Another approach is to check previously published ERAD substrates known to have charged residues in the transmembrane domains. The best-characterized example is the TCRalpha chain, which carries two conserved lysines in its transmembrane domain. The authors mentioned it in the discussion and suggested that the degradation of TCRalpha is not dependent on the Membralin complex. They should include this data and discuss possible models to explain how the Membralin complex may select specific substrates but ignore others with a similar lesion.

1- Please see our response to Reviewer #1, point 5 that raised similar issues.

2. Another central open question is how a quality control ligase recognizes substrates

bearing a charged residue in a transmembrane domain. Fully addressing this question is undoubtedly beyond the scope of this study. However, the authors can include more experiments to bolster the concept that the Membralin complex can recognize an unassembled membrane protein with a charged residue in the transmembrane domain. For example, does the ligase complex also degrade proteins bearing negatively charged residues in the membrane? Which subunit of the complex is responsible for substrate recognition? Is substrate recognition mediated by the transmembrane domains of a ligase subunit? Is there any unique conserved sequence feature in the ligase that may allow such recognition?

2- We fully agree with the reviewer that this is an important question and that it exceeds the scope of this manuscript. We have included an experiment that showing that in cells lacking RNF185, TMUB1 and TMUB2, MBRL is capable of interacting with TPSN (Figure 3f). These data suggest that among the components of this ERAD complex, MBRL can act in substrate recruitment/recognition. Our previous paper reporting the discovery and initial characterization of the RNF185/MBRL ERAD complex also included co-immunoprecipitation experiments suggesting that among the various subunits, MBRL is the one that interacts more strongly with substrates. Whether the interaction is direct and the precise features recognized on substrates should be clarified in future studies.

3. The authors show that in the absence of the Membrane complex, more unassembled Tapasin is present in the cell. Coincidentally, more MHC molecules are found on the cell surface. Are these two observations functionally linked? If so, why can more unassembled Tapasin help more MHC reach the cell surface? The experiment in Figure 5 suggests a possibility that contradicts the proposed quality control function of the Membralin complex. This ligase seems to degrade Tapasin and TAP1/TAP2 after these proteins have assembled into a functional complex. Can the authors design an experiment to elucidate whether the Membralin complex targets unassembled Tapasin during PLC biogenesis or mature PLC after the complex is disassembled (which may be an integral regulatory step in peptide loading)? If it is the latter case, it will deviate the study from the previously reported paradigm of membrane protein quality control and, thus, significantly improve the study's novelty.

3- Previous studies showed that the steady state levels of TPSN, TAP1 and TAP2 are interdependent (PMID: 33275281). In agreement with those observations, we show that depletion of TAP1 results in low TPSN levels and that these can be restored by additional deletion of RNF185/MBRL. These data suggested that TPSN assembly with TAP1/2 and degradation by RNF185/MBRL are competing events. According to this model, deletion of RNF185/MBRL would shift the balance towards TPSN assembly with TAP1/2 resulting in higher levels of these proteins, as shown in Figure 5. We hypothesized that an excess of TAP1 would also shift the balance towards TPSN assembly in cells with an active RNF185/MBRL complex. Indeed, we observe that TAP1 overexpression results in an increase of both the steady state levels and the stability of TPSN (Figure 4b, c). Altogether these data argue in favour that RNF185/MBRL degrade unassembled TPSN while when assembled, TPSN is not recognized by this ERAD

complex. As an aside, this last experiment showed that turnover of overexpressed TAP1 is relatively slow and independent of MBRL.

4. Is the degradation defect of PLC relevant to the in vivo Membranlin knockout mouse phenotype? Do the Membranlin knockout astrocytes show signs of inflammation or astrogliosis that can be suppressed by knocking down the PLC. The authors may use qRT-PCR to check the expression of well-characterized cytokines and chemokines involved in astrocyte activation. Results along this line may explain the observed CNS dysfunction and animal death phenotypes.

4- Earlier characterization of astrocyte specific MBRL KO mice showed that these animals develop strong neuroinflammation (PMID: 31112137). Transcriptomics analysis showed a signature characteristic of reactive astrocytes with upregulation of pro-inflammatory cytokines such as TNF-alpha. We agree with the reviewer that it will be interesting to test whether these features can be suppressed by depleting PLC components in RNF185/MBRL-deficient mice. However, we believe that these studies in mice fall outside the scope of the current work and should be addressed in the future.

Minor points:

The authors should include some subheadings to improve the readability of the result section.

Subheadings have been included to improve readability.

Reviewer #4 (Remarks to the Author)

The manuscript by van de Weijer et al describes an ER associated degradative (ERAD) pathway that acts on tapasin to regulate levels of cell surface MHC-I. The three components of the pathway and a selection of substrates were previously identified by the authors in astrocytes and the present manuscript extends these earlier observations to tapasin. An unbiased proteomics screen on ERAD KO cells showed enrichment for tapasin implicating this degradative pathway in controlling steady state tapasin levels. The pathway is shown to function not only in astrocytes but also in a variety of conventional cell lines. Interestingly this ERAD pathway selectively targets only tapasin among the members of the peptide loading complex (PLC). The authors identify conserved a lysine in the transmembrane domain of tapasin as crucial for targeting by the ERAD pathway. The same lysine is also necessary for interaction with the TAP transporter for proper PLC functioning. Overall the experiments are well carried out, the data clearly presented, and the writing is concise. I have only a few minor points:

1. In the second para of the Results section: “restored MHC-I antigen presentation” is not strictly correct as no assay was performed for presentation, e.g. with a T cell read out. Instead “restored surface levels of MHC-I” would be the more appropriate phrasing.

We agree with the reviewer and modified the text to more accurately describe our findings.

2. Can the authors speculate on the efficiency of this ERAD pathway in different tissues? Since surface MHC-I levels are known to vary among different cell types, could these differences be ascribed partly to the upregulation or downregulation of components of the ERAD pathway.

Our data show that TPSN is a substrate of the RNF185/MBRL complex in cells from diverse origins and organisms. However, the efficiency of this process across different tissues is difficult to assess and likely to depend on multiple factors. The levels of the RNF185/MBRL certainly play an important role but the relative expression of the various PLC components is also important (as suggested by the experiments with depleted and overexpressed TAP1). Future studies using more physiological systems will likely provide important insights into the process.

3. In Methods under the subheading “Proteome sample preparation...”, the first line states “A total of 50 ug protein was reduced...”. However I could not find a description of how cell lysates were prepared for this step. Were the lysates prepared using 1% DMNG as described under “Co-immunoprecipitation”? Also, that subheading should be “from ex vivo mouse astrocyte culture” not “to ex vivo mouse astrocyte culture”.

Mouse astrocytes were lysed in 5% SDS 50mM TEAB lysis buffer (pH 7.55). This information was now added to the Material and Methods section.

REVIEWERS' COMMENTS

Reviewer #1 (Remarks to the Author):

The authors have now appropriately answered our questions

Reviewer #2 (Remarks to the Author):

Reviewer #3 (Remarks to the Author):

The authors have addressed all my criticisms. I support the publication of this paper in Nat. Communications.

Reviewer #4 (Remarks to the Author):

The authors have satisfactorily addressed my concerns and I have no objection to publication.